# Learning to localize sounds in a highly reverberant environment: Machine-learning tracking of dolphin whistle-like sounds in a pool

**Sean F. Woodward**[1]\*, **Diana Reiss**[2], **Marcelo O. Magnasco**[1]

**1** Laboratory of Integrative Neuroscience, Center for Studies in Physics and Biology, The Rockefeller University, New York, NY, United States of America, **2** Department of Psychology, Hunter College, City University of New York, New York, NY, United States of America

\* sf.woodward@gmail.com

**Data Availability Statement:** Data used for model building and testing available at https://doi.org/10.6084/m9.figshare.7956212.

## Abstract

Tracking the origin of propagating wave signals in an environment with complex reflective surfaces is, in its full generality, a nearly intractable problem which has engendered multiple domain-specific literatures. We posit that, if the environment and sensor geometries are fixed, machine learning algorithms can "learn" the acoustical geometry of the environment and accurately track signal origin. In this paper, we propose the first machine-learning-based approach to identifying the source locations of semi-stationary, tonal, dolphin-whistle-like sounds in a highly reverberant space, specifically a half-cylindrical dolphin pool. Our algorithm works by supplying a learning network with an overabundance of location "clues", which are then selected under supervised training for their ability to discriminate source location in this particular environment. More specifically, we deliver estimated time-difference-of-arrivals (TDOA's) and normalized cross-correlation values computed from pairs of hydrophone signals to a random forest model for high-feature-volume classification and feature selection, and subsequently deliver the selected features into linear discriminant analysis, linear and quadratic Support Vector Machine (SVM), and Gaussian process models. Based on data from 14 sound source locations and 16 hydrophones, our classification models yielded perfect accuracy at predicting novel sound source locations. Our regression models yielded better accuracy than the established Steered-Response Power (SRP) method when all training data were used, and comparable accuracy along the pool surface when deprived of training data at testing sites; our methods additionally boast improved computation time and the potential for superior localization accuracy in all dimensions with more training data. Because of the generality of our method we argue it may be useful in a much wider variety of contexts.

## Introduction

Principled methods to track the spatial origin of sounds using a microphone array rely on differences in the time of arrival and the intensity of the sound as registered across array

**Funding:** MO Magnasco, DR Reiss Awards 1530544, 1607280 National Science Foundation https://www.nsf.gov The funders had no role in study design, data collection and analysis, decision to publish, or preparation of the manuscript.

**Competing interests:** The authors have declared that no competing interests exist.

elements. This is subject to a central assumption: that the direct, straight-line-path arrival of the sound at each element can be computationally identified and isolated from the raw waveforms recorded in the array. Acoustically-reflective surfaces create echoes which arrive at the destination through different paths; the pattern of differences in time of arrival and intensities may make the echoes appear to originate from locations other than the source, just like optical reflections in a mirror appear to originate in a place other than the source, called a *virtual image*, as shown in Fig 1. When sound propagates through multiple reflections before being damped, we have a reverberant environment, where it may be impossible to computationally distinguish the straight-line-path incidence from the echoes. Particularly for tonal sounds, the amplitude envelope may be distorted differentially across different array elements due to the echoes arriving in a different temporal order at different array elements, as shown in Fig 1B. At this point, principled methods are unable to track. This problem has also been studied in the radio wave propagation literature, where it goes by the name of "multipath", being particularly relevant to GPS and cell signal propagation in cities with dense high-rises.

In this paper we consider the case of tracking tonal dolphin whistles in an aquarium pool with extensive reverberation. Underwater sound propagation has unique characteristics which distinguish it from aerial sound localization: sound travels five times faster in water than in air, it is subject to refraction at thermoclines and haloclines, it has longer propagation distance at many frequencies, and the the water-air interface presents a consistent overhead reflective and scattering surface. On the other side, the 1.5 km/s speed of sound, together with the slower

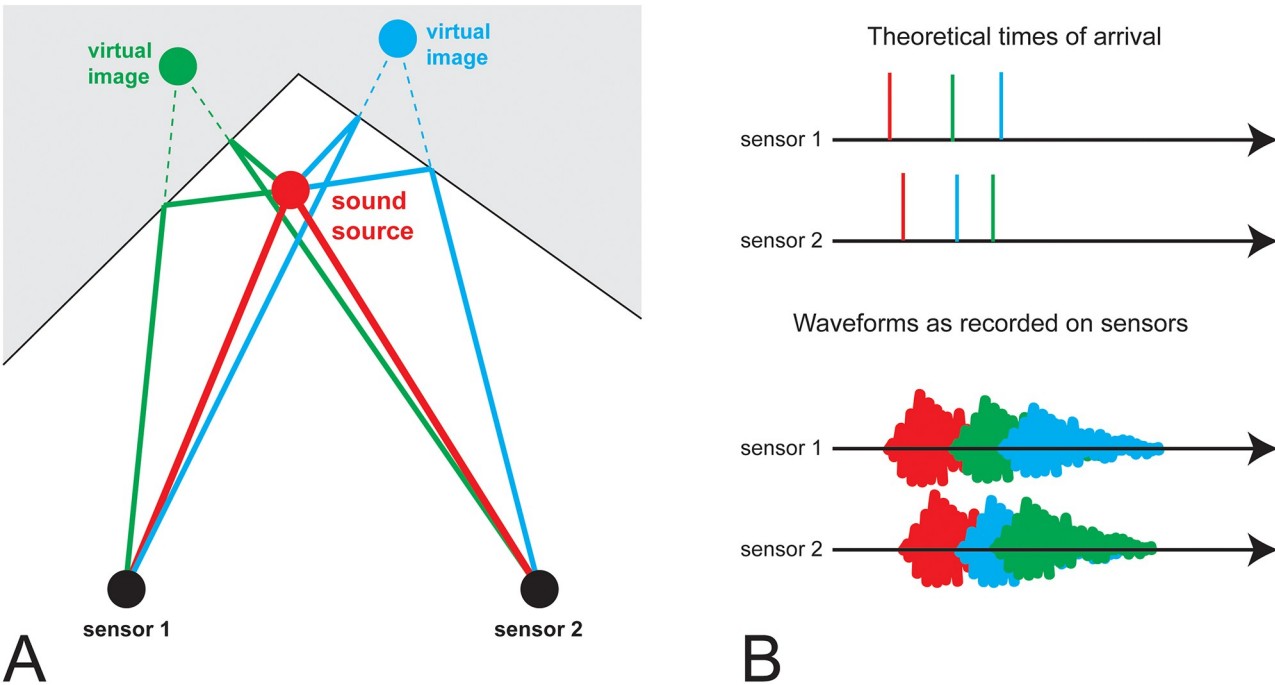

**Fig 1. Cartoon depiction of a sound's arrival at sensors in a reflective environment.** (A) Sound originates from a source and arrives at two sensors, both directly (red paths) or bouncing on walls (green and blue). The sound bouncing off the walls arrives at the sensors at angles and delay times matching what would have happened if the sound had originated at a "virtual image", located at the mirror reflection of the source on the corresponding wall. (B) The arrival times of the sounds at each sensor; because the source is slightly to the left the direct incidence (red) arrives just slightly earlier at sensor 1; because the left image (green) is to the left that echo arrives much earlier at sensor 1 than 2, and because the right image (blue) is to the right, that echo arrives much earlier at sensor 2 than 1. If the sounds were purely impulsive then the first incidence would always be clear, but matching later echoes requires computation. However, if the sounds are longer than the typical distance between echoes, then the recorded waveform contains overlapping pieces, and any particular method of computing TOA (e.g. by cross-correlation) may give erroneous results due to the echoes arriving in a different temporal order at each sensor.

speed of swimming animals than, say, flying animals, conspire to make Doppler shift a lesser concern.

As we shall discuss in far more detail below, dolphins produce an array of different vocalization types, both impulsive and tonal. The tonal sounds, termed *whistles*, that dolphins produce are generally in the 4-25 kHz range [1] and are hard to track in reverberant environments because multiple echoes may arrive at any given hydrophone within the duration of the sound itself, and these echoes may arrive in a different order at different microphones, making time delay tracking inviable, as shown in Fig 1.

However, for a situation where a fixed hydrophone array has been placed in a fixed environment, we could imagine that the geometric idiosyncrasies of the environment could be mapped out. Particularly with several hydrophones, one can build a lot of information which in principle is redundant, but which in a particular environment may have vastly variable degrees of reliability. It should be in principle possible to choose the right combination that permits localization in this particular environment; the question is what is the best way to do so.

Our proposal is in principle simple. *First*, we created a "training dataset", by placing an emitter in a number of known locations within the aquarium pool, and playing back through the emitter a repertoire of computer-generated sounds approximating dolphin whistles; over 1700 of these sounds were recorded using our 16-hydrophone array deployed at the boundary of the pool. The original signal that was played through the emitter is, of course, never again used. The 16-track, 2-second recording of each signal and its echoes is a datum in our dataset, and is associated with a label denoting the source location. *Second*, for each datum we compute an overabundance of "clues"; for example, we use several distinct cross-correlation methods to find time difference estimates $\Delta t$ between every pair of channels, of which there are of course $16^*15/2 = 120$ pairs.

What is the logic in doing so? To calculate the 3D location of the source by triangulation, a minimum of 4 reliable $\Delta t$ are required, since in addition to the $(x, y, z)$ coordinates we need the time at which the sound originated at the source. In order to use more data than 4 $\Delta t$s we can set up an overdetermined system and solve it in the least squares sense. In a non-reverberant environment this is useful since $\Delta t$ estimates have some imprecision or temporal jitter; obviously, the smaller the jitter, the smaller the $\Delta x$ imprecision in the estimation of the sound source coordinates, but also this imprecision is lowered if there are more $\Delta t$ available. However, in a reverberant environment, many of those clues will be **wrong**, because any given estimator of $\Delta t$ may try to match a first arrival in one channel to an echo in another channel, yielding a $\Delta t$ estimate which is, we emphasize, not inaccurate but incorrect. Due to the position of the hydrophones with respect to the pool walls, and the way the reverberation echoes arrive at those particular locations, it may be the case that some comparisons are often incorrect for many sounds. Inputting such potentially misleading data into a least squares algorithm risks corrupting the whole estimate. *By computing a large number of different estimates we make sure we have some that are often correct*. Learning is then a matter of sorting the wheat from the chaff.

So, after computing our overabundance of estimates, then *third*, we use a supervised learning method to train a classifier to predict, from these many clues, the label. Any regularization that promotes sparseness will rapidly discard the cross-comparisons that are often misleading. In doing so, the machine learning method is, de facto, learning how to track the sounds *in this particular acoustical environment*. If the environment or the hydrophone position were changed, the good clues would change. We shall show below that this strategy performs rather well.

*Fourth*, finally, once the system has learned to classify, *running* the system for novel data only entails computing those specific clues that were selected by learning, and running forward the classifier. This is computationally a rather lightweight operation in comparison with training, and it can be performed online in real time.

The outline of the rest of this Paper is as follows. In *Background and Significance* we shall review both the motivation to study these particular bioacoustical signals, as well as an extremely summary review of the extensive amount of work that has been done in the field of marine mammal acoustic localization. In *Materials and Methods* we shall review our hydrophone array design and layout, the creation of the sounds used, playback and acquisition methods, features, and machine learning methodology. Then we list our *Results* and discuss them in *Discussion*, and finally speculate on generalizations in *Conclusions and Outlook*.

## Background and significance

Dolphin communication research is in an active period of growth. Many researchers expect to find significant communicative capacity in dolphins given their complex social structure [1–3], advanced cognition including the capacity for mirror self-recognition [4], culturally transmitted tool-use and other behaviors [5], varied and adaptive foraging strategies [6], and their capacity for metacognition [7]. Moreover, given dolphins' well-studied acoustic sensitivity and echolocation ability [8–10], some researchers have speculated that dolphin vocal communication might share properties with human languages [11–13]. However, there is an insufficiency of work in this area to make substantive comparisons.

Among most dolphin species, a particular narrowband class of call, termed the *whistle*, has been identified as socially important. In particular, for the common bottlenose dolphin, *Tursiops truncatus*—arguably the focal species of most dolphin cognitive and communication research—research has focused on *signature whistles*, individually distinctive whistles [14–16] that may convey an individual's identity to conspecifics [15, 17] and that can be mimicked, potentially to gain conspecifics' attention [18].

Signature whistle studies aside, most studies of bottlenose dolphin calls concern group-wide repertoires of whistles and other, pulse-form call types [19–23]; there is a paucity of studies that seek to examine individual repertoires of non-signature whistles or the phenomenon of non-signature acoustic exchanges among dolphins. Regarding the latter, difficulties with whistle source attribution at best allow for sparse sampling of exchanges [17, 24]. Nevertheless, such studies constitute a logical prerequisite to an understanding of the communicative potential of whistles.

The scarcity of such studies can be explained in part by limitations in available Passive Acoustic Monitoring (PAM) analyses, in particular for the source attribution of whistles to freely-interacting members of a social group. A version of the sound attribution problem is encountered in many areas of passive acoustic research, often representing a necessary step to making source comparisons. We provide a brief overview of sound source attribution in the context of marine mammalogy, with a focus on the present application.

Sound source attribution is closely related to sound source separation, the separation of audio data composed of sounds (such as calls) from many sources into individual, source-specific tracks, which is a common task in marine mammalogy. Sound separation might be accomplished by recognizing sound features (durations, repetitions, frequency components, frequency component modulations, etc.) that are distinctive to sources and that imply separations. Examples include the separation of dolphin signature whistles from non-signature whistles based on differences in repetition characteristics [24], and the separation of calls belonging to different marine mammal species [25] or different intra-species regional groups, such as

groups of sperm whales [26], based on differences in call spectrographic characteristics. However, if sound sources are not known to generate inherently distinct sounds, as is the case when focusing on bottlenose non-signature whistles, sound separation might be performed on the basis of identifying their distinctive source locations. Often this is achieved by obtaining explicit source locations or coordinates for the sounds of interest, which is called sound source localization.

At this point, we distinguish sound source separation from sound source attribution. In marine mammalogy, the former often carries the connotation that sound sources are separated exclusively on the basis of audio data. When this is done using sound source localization, source attribution is often implied by the sustained large spacings among oceanic individuals/groups [27] or by an emphasis on regions as sources rather than individuals/groups, as in abundance surveys [25, 28, 29]. In other words, in these cases a source's location identifies the source. Bt contrast, aquarium dolphins intermix rapidly as compared with their whistle rate, such that a sound's location does not uniquely identify the source. For this case, we refer to sound attribution as the the process of not only separating a sound by localization, but individually attributing it to a particular dolphin based on the dolphin's visual coordinates and identifiers, obtained from accompanying video feed.

Whether the end goal is sound source separation or explicit attribution, the fundamentals of sound source localization are the same. Sound source localization refers to locating the spatial origin of a sound based on an understanding of predictable phenomena that affect its waveform between its production at the source and its receipt at (hydro)phones, from which acoustic data are taken. Such phenomena include but are not necessarily limited to intensity attenuation, dispersion, and time delay. The last is the focus of many marine mammal call localization methods. Because a sound's travel distance is the mathematical product of its travel time and the speed of sound, time delay information implies distance information about a sound source with respect to the hydrophones. Because in general we do not know the absolute travel time of a sound between source and hydrophone (given that the time of sound production is unknown), we are often interested in time-difference-of-arrivals (TDOA's), the differences in a sound's time-of-arrivals (TOA) within pairs of hydrophones. Several methods are available for solving the optimization problem of localizing a sound based on a set of estimated TDOA's for several hydrophones (typically four or more) [30–32], and in this paper with primarily refer to one, Spherical Interpolation. This method has been proven to be optimal if the error in TDOA estimates is Gaussian [33]. Alternatively, some sound localization methods (namely beamforming methods) are based on TDOA estimations that are not explicitly obtained [34–37], though ultimately these rely on similar techniques of sound processing as discussed below.

Much of the difficulty in sound source localization lies in obtaining reliable TDOA estimates from the waveforms of received sounds. For broadband, impulse-like sounds (characterized by fast, high-intensity, sparse onsets) in certain environments, this can be as straightforward as estimating arrival times to be the peaks in the waveforms [38]. More frequently, one seeks to leverage information from the whole waveforms. Traditionally, this can be done by using the cross-correlation or one of its close mathematical relatives. In essence, the cross-correlation computes the dot-product of two waveforms across a range of relative shifts in time. Assuming the sound waveforms received by two hydrophones are identical apart from their arrival times, their cross-correlation will reach a maximum at the relative time shift corresponding to their TDOA. This assumption of identical, shifted waveforms is valid enough to accurately estimate TDOA's, and in turn perform sound source localization, for certain sounds (particularly impulse-type), in large and low-reverberation recording environments, and/or

when localization precision can be relatively low. These conditions often co-occur and apply to oceanic survey studies seeking to achieve source separation versus attribution [39–46].

However, the assumption of identical, shifted received waveforms is never strictly correct, in large part because of the effects of multipath. Multipath refers to a sound taking multiple paths through space between its source and each hydrophone, including both the direct path as well as paths involving reflections from nearby surfaces. As a result, each hydrophone receives multiple imperfect copies of the source waveform (imagine your ears receiving echoes in an opera house), rather just one perfect copy. In an enclosed environment (which induces substantial multipath), copies that arrive within 50 ms are termed reverberations. If the original sound was a short-duration impulse, such that a single waveform copy fully arrives in a hydrophone before the next starts, it may be possible to undo multipath effects. In general, however, the result of cross-correlating two hydrophones' samples of a sound subject to multipath is a data series with multiple local peaks ($N^2$ peaks given that waveforms from N paths arrive in each hydrophone) and no easy way to determine the peak corresponding to the desired direct-path TDOA.

Various strategies have been proposed to make the cross-correlation robust to multipath effects [47, 48], such as by locating the correct peak in the cross-correlation of two sets of stacked waveforms [49]. More recent methods have been proposed for sperm whale click trains, using statistical approaches to reconcile multiple sets of estimated TDOA's (including direct-path and higher-order paths) obtained for clustered impulse sounds [50, 51]. While generally promising, it remains unclear to what extent these methods generalize to longer-duration narrowband sounds, particularly in reverberant environments. These methods belong to a larger class of methods targeted at impulse-like sounds such as sperm whale clicks or to narrowband sounds produced in large non-reverberant environments (where reflections are sparse), in which direct-path and secondary-path waveforms are separable and amenable to geometric interpretation [27, 52–54]. Perhaps more pertinent to narrowband sounds in reverberant environments is a method recently proposed for transforming complex waveforms into impulse-like waveforms (impulse response functions) for easier TDOA extraction [55], however at present this method has not been tested for marine mammal call localization in any context.

For the task of localizing dolphin whistles in particular, standard cross-correlation methods have typically been applied, with at best modest results in the relatively irregular, low-reverberation environments where they have been evaluated [44, 56–58]. The method with the best proven ability to localize dolphin whistles in a reverberant environment to date falls under the umbrella of Steered-Response Power (SRP). In short, these methods rely on the finding the spatial coordinates that best explain the received signals, under the assumption that cross-correlating (and summing) the received signals shifted by the set of TDOA's implied by those spatial coordinates' will result in a numerical maximum [59]—this will be discussed more in the Tonal Localization subsection of our Materials and Methods. Rebecca E. Thomas et al. [60] have demonstrated the use of such a method with bottlenose whistles in an enclosed environment with reasonable success (used for about 40% recall of caller identity) [60].

We note that an alternative solution to whistle source attribution that does not involve sound source localization is the use of sound transducers attached to the dolphins themselves [61–63]. While promising, shortfalls include the need to manually tag every member of the group under consideration, the tendency of tags to fall off, and the tags' inherent lack of convenient means for visualizing caller behavior. Most significant to research with captive dolphins, the use of tags can conflict with best husbandry practices (e.g., due to risk of skin irritation, of ingestion) and be forbidden, as is the case at the National Aquarium. At such locations, less invasive means of sound source attribution are necessary.

In this paper, we propose methods of source sound localization suited for sound source attribution that offer good results for long-duration, whistle-like sounds in a highly reverberant marine environment, the half-cylindrical artificial dolphin pool located at the National Aquarium in Baltimore, Maryland. Our methods, taken from machine learning, are distinct from those previously discussed in that they do not approach the problem of multipath by explicitly modeling it in any way, and in that a majority of computation is done before a sound of interest is to be localized. The disadvantage of these methods is the need to prepare them by playing numerous calibration tones in a space of interest, suiting them to small enclosures.

Our raw data consist of a set of artificial, frequency-modulated, narrowband sounds (referred to as "tonals" here) that were designed to broadly reflect the scope of *T. truncatus* whistles. The data included the waveforms received by 16 hydrophones, separated among four 4-hydrophone-arrays surrounding the pool, after tonal play at several known source locations inside the pool. We first show that a nonlinear random forest "classification model" (i.e., a predictor of categorical group membership) succeeds at correctly choosing a played tonal's source location from the set of all source locations. Later, we show that a linear classification model achieves similar results. Finally, we show that regression models can predict the two or three dimensional source coordinates of played tonals with dolphin-length accuracy (if not inter-dolphin-head accuracy, which approximates minimum possible source separation), depending on whether the models were built with data from source locations that were also to be predicted. The latter two models rely on fewer than 10,000 values to predict each played tonal source location. We implement an established SRP method for evaluation of our source localization regression models, obtaining favorable results along the pool surface but not in the direction of pool depth. Finally, reducing our feature set even further by building a parsimonious (minimum-feature) classification tree for the same task, we find that a minimally sufficient feature set for classification includes data from all pairs of 4-hydrophone-arrays, consistent with the data valued by a strictly geometric, TDOA-based approach to sound source localization.

## Materials and methods

### Hydrophone setup

All data were obtained from equipment deployed at the Dolphin Discovery exhibit of the National Aquarium in Baltimore, Maryland. The exhibit's 33.5-m-diameter cylindrical pool is subdivided into one approximate half cylinder, termed the *exhibit pool* (or EP, Fig 2A), as well three smaller holding pools, by thick concrete walls and 1.83 m x 1.30 m perforated wooden gates; all pools are acoustically linked. The acoustic data were obtained from the EP, when the seven resident dolphins were in the holding pools; their natural sounds were present in recordings.

Sound data were collected by 16 hydrophones (SQ-26-08's from Cetacean Research Technology, with approximately flat frequency responses between 200 and 25,000 Hz) placed inside the EP. The details of our hydrophone placement were constrained in large part by National Aquarium operational concerns. Within these constraints, we chose to split the 16 hydrophones among four 4-hydrophone-arrays designed for long-term deployment, which we splayed about the EP as highlighted by the yellow circles in Fig 2A. We chose to place four hydrophones in each array primarily for redundancy, secondarily to accommodate beamforming methods of sound source localization. We chose to evenly distribute the four 4-hydrophone-arrays along the curved EP wall to approximately maximize the time-difference-of-arrivals (TDOA's) among the four hydrophone clusters across all potential source locations

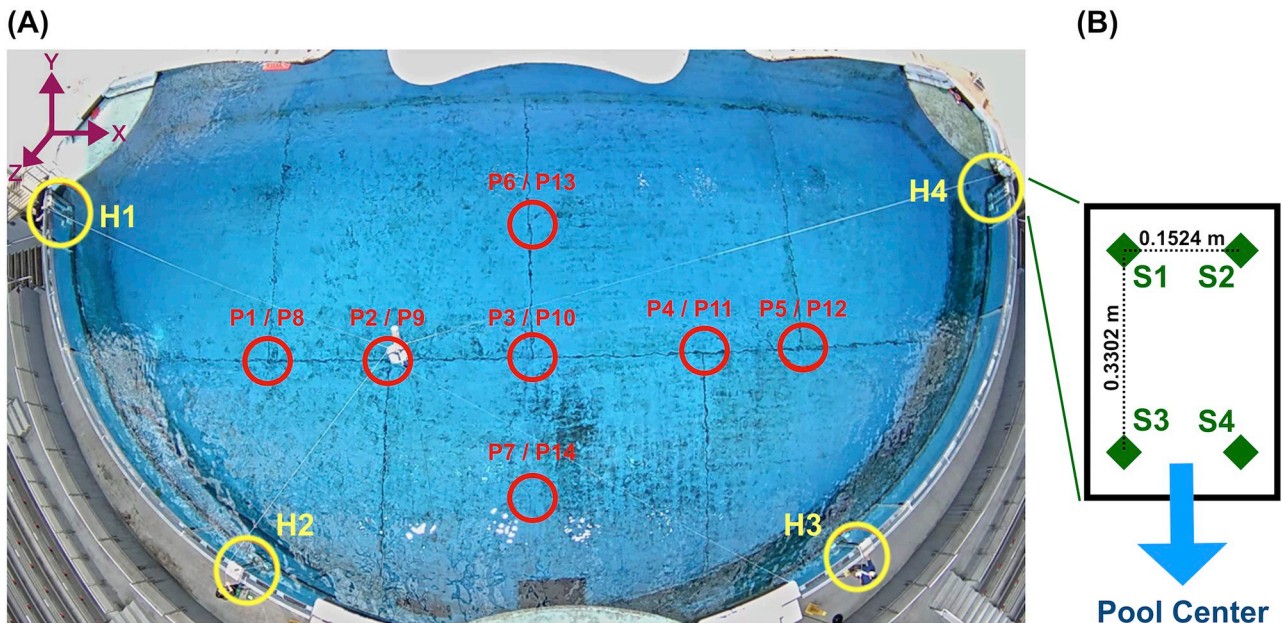

**Fig 2. Layout of the tonal source locations and the four 4-hydrophone-arrays.** (A) The National Aquarium Exhibit Pool (EP) is shown, as visualized by an overhead AXIS P1435-LE camera. Circled in red are the approximate surface projections of the 14 source locations at which tonals were played; each circle represents both a shallow and deep source location, corresponding to the two forward-slash-separated identifiers adjacent each circle. Circled in yellow are the four 4-hydrophone arrays, with adjacent identifiers. (B) The simplified face of a hydrophone-containing panel from one 4-hydrophone-array. Green diamonds represent approximate hydrophone positions, with local addresses adjacent.

inside the EP. This has been shown to be helpful to TDOA-based methods of sound source localization [64–66].

Moving from left to right in Fig 2A, we refer to the 4-hydrophone-arrays as H1, H2, H3, and H4. The functional component of each 4-hydrophone-array was a 0.3058 m x 0.4572 m x 0.0762 m acrylic panel, which contained one locally-addressed hydrophone near each corner (Fig 2B) and was cushioned against the acrylic pool wall 1.6000 m (measuring from the panel center) below the water surface. The origin of our three-dimensional Cartesian coordinate system was located at the center of H1's acrylic panel. The coordinates of the 16 hydrophones are given in Table 1. We manually measured the coordinates of the individual hydrophones using both rules and laser rangefinders, and demonstrated that these hydrophone coordinates accommodated localization of calibration signals (using the geometric, TDOA-based method of sound source localization termed Spherical Interpolation [31, 33, 67]) with error < 1 m. While we did not require precise coordinates for the vertical pool walls for any purpose, for visualization we approximated these walls as located on the closed bottom half of a 33.5-m-diameter circle centered at (15.7 m, 3.87 m).

## Tonal creation

The sound data collected by our hydrophones consisted of computer-generated, whistle-like tonals played over an underwater speaker (a Lubbell LL916H). This choice of sound data, intended to approximate whistle data collected from bottlenose dolphins, was motivated by a few considerations. First, we required that the tonals were not previously subject to multipath phenomena (i.e., were not pre-recorded), so as not to risk skewing our source localization models with false source information. Second, we required that the tonals were broadly representative of the approximate "whistle space" for *Tursiops truncatus*, or distributed over a

**Table 1. Hydrophone coordinates.**

| Hydrophone Number | Inter-Intra Array Address | X (m) | Y (m) | Z (m) |
|---|---|---|---|---|
| 1 | H1-S1 | 0.0059 | -0.0760 | 0.1905 |
| 2 | H1-S2 | -0.0059 | 0.0760 | 0.1905 |
| 3 | H1-S3 | 0.0059 | -0.0760 | -0.1905 |
| 4 | H1-S4 | -0.0059 | 0.0760 | -0.1905 |
| 5 | H2-S1 | 9.1521 | -10.5672 | 0.1905 |
| 6 | H2-S2 | 9.0089 | -10.5148 | 0.1905 |
| 7 | H2-S3 | 9.1521 | -10.5672 | -0.1905 |
| 8 | H2-S4 | 9.0089 | -10.5148 | -0.1905 |
| 9 | H3-S1 | 23.2718 | -10.3148 | 0.1905 |
| 10 | H3-S2 | 23.1288 | -10.3672 | 0.1905 |
| 11 | H3-S3 | 23.2718 | -10.3148 | -0.1905 |
| 12 | H3-S4 | 23.1288 | -10.3672 | -0.1905 |
| 13 | H4-S1 | 31.7886 | 0.2984 | 0.1905 |
| 14 | H4-S2 | 31.7768 | 0.1464 | 0.1905 |
| 15 | H4-S3 | 31.7886 | 0.2984 | -0.1905 |
| 16 | H4-S4 | 31.7768 | 0.1464 | -0.1905 |

Refer to Fig 2 for Inter-Intra Array Address orientation.

generous range of relevant whistle characteristics. These first two considerations complicated the use of real whistle playbacks. Lastly, we desired that the tonals were obtained in sufficient quantity (on the order of hundreds of tonals per classification group, consistent with typical machine learning sample sizes), from well-measured source locations inside the EP. While this would not have been impossible using whistles produced by the pools' live dolphins, at this stage of study we hoped to isolate sound source localization from the separate task of visual object localization (i.e., obtaining dolphin coordinates from cameras), which this method would have conflated; a semi-stationary speaker could be more reliably measured.

Another choice we made during tonal creation and subsequent source localization model creation was to focus on the "fundamental" (lowest-frequency component) of the dolphin whistle, ignoring the accompanying set of harmonics, or components that are "stacked" above the fundamental in frequency. This choice was made to avoid making perilous and unnecessary assumptions about the mathematical relationship between fundamentals and harmonics during signal creation, and to avoid expanding the size of our target whistle space by adding degrees of freedom. Above all, this choice was made because the whistle fundamental is generally understood to be the strongest-intensity whistle component as well as separable from the harmonics as a result of their signal-spanning displacements in frequency [68, 69]. Thus, with appropriate filtering, the problem of localizing the source of a whistle should be reducible to the problem of localizing its fundamental, with the excluded harmonics representing additional information that we would only expect to improve the quality of localization (were we to overcome the drawbacks). Moreover, by focusing on fundamentals, we hoped our methods would be more applicable to localizing two or more whistles produced simultaneously by two or more sources; the methods only require that the whistles' fundamentals, and not their harmonics, be cleanly filtered of overlaps in time-frequency space.

We created 128 unique tonals with pitch, duration and other parameters spread across published ranges of *T. truncatus* whistles [69, 70]—we note that these parameters assume whistles with sinusoid characteristics. Our chosen tonal parameter values are given in Table 2. Two

**Table 2. Parameter values of created tonals.**

| Parameter | Value Set |
|---|---|
| Duration (sec) | [0.3, 1] |
| Number of Cycles | [1, 2] |
| Center Frequency (Hz) | [6000, 10500] |
| Cycle Amplitude (Hz) | [2000, 5000] |
| Phase Start (rad) | $[-\frac{\pi}{2}, \frac{\pi}{2}]$ |
| Power Onset/Decay Fraction * | [0.1, 0.25] |

* Refer to the body for an explanation of "Power Onset/Decay Fraction."

tonals were created for each permutation of table parameter values, as described below. To construct a tonal's waveform, we began with an instantaneous frequency, $f(t)$, that described a goal time-frequency (or spectrographic) trace, for instance the trace shown in Fig 3. For simplicity, and consistent with the parameters typically used to describe dolphin whistles, we approximated dolphin whistles as sinusoidal traces in spectrographic space—thus $f(t)$ was always a sinusoid. Based on the standard definition of the instantaneous frequency as

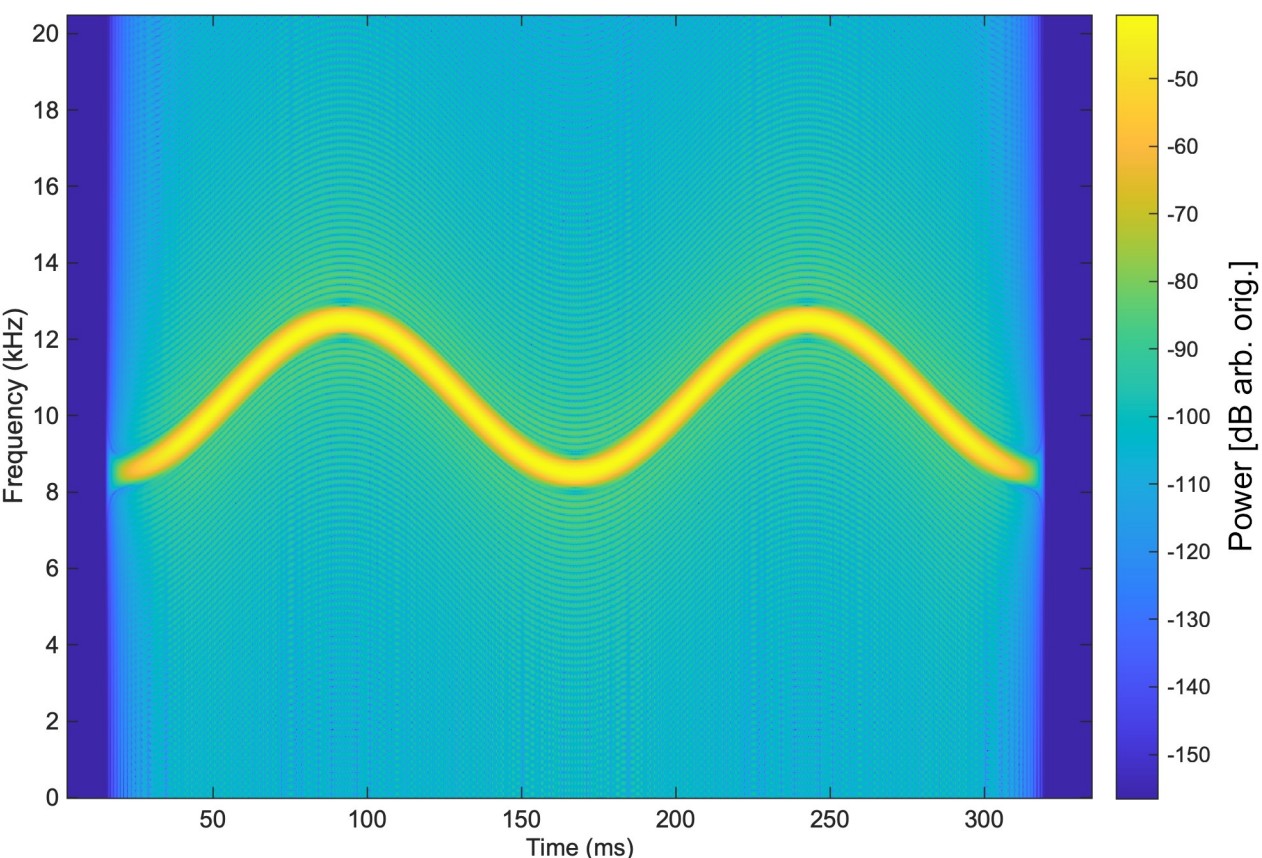

**Fig 3. Spectrogram of an artificial whistle.** Displayed is a standard, 1024-bin, Hamming-window spectrogram of one of the 128 tonals that was played (and here sampled) at 192 kHz; frequency resolution of the plot is 187.5 Hz (smoothing added). Note that the spectrogram was constructed from the unplayed, computational source signal (power level is arbitray). Duration = 0.3 s, Number of Cycles = 2, Center Frequency = 10500 Hertz, Cycle Amplitude = 2000 Hertz, Phase Start = -π/2.

$f(t) = \frac{1}{2\pi}\frac{d\Phi(t)}{dt}$, we obtained the phase $\Phi(t)$ by integration of $f(t)$ with respect to time. The phase could be straightforwardly transformed into a playable waveform $y(t)$ as $y(t) = A(t)sin(\Phi(t) + \alpha)$, where $\alpha$ represents "Phase Start" from Table 2 and $A(t)$ denotes a piecewise function that enforced a gradual onset and decay of tonal intensity. Given a tonal length of $X$, a "Power/Decay Onset Fraction" of $P$ (see Table 2), and a tonal onset time of $t = 0$, the component of $A(t)$ responsible for a gradual onset would take the form $sin^2\left(t\frac{2\pi}{4XP}\right)$, acting between $t = 0$ and $t = XP$; the decay component of $A(t)$ would be a mirrored version, acting at the end of the tonal. Instead of creating the signal $y(t)$ as just described, the phase derived for $f(t)$ could be transformed with a heuristic into a waveform corresponding to a slightly modified version of $f(t)$, specifically $y(t) = A(t)\frac{arcsin(m \cdot sin(\Phi(t)))}{arcsin(m)}$. $m$ is a nonzero parameter less than one that renders the underlying waveform more triangular, with the effect of creating a harmonic stack in time-frequency space (each harmonic successively decreasing in intensity as a function of total harmonics), as the parameter decreases from one. We used a value of $m = 0.8$, for which four harmonics were created; in practice, we rarely observed the second harmonic (and never the third onward), which is ~50 dB less in signal power than the fundamental, above noise in the pool.

## Tonal play and acquisition

Using a Lubbell LL916H underwater speaker, which possesses an approximately omnidirectional sound profile, the 128 created tonals were played at calibrated volume levels at each of 14 source locations inside the EP, corresponding to 7 surface positions and two depths, 1.60 m and 5.87 m below the water surface (Fig 2A, Table 3). Surface spacing between adjacent source locations was approximately 1.5 m—5 m. The speaker was suspended by rope from a custom flotation device and moved across the pool surface by four additional ropes extending from the device to research assistants standing on ladders poolside. Importantly, the speaker was permitted to sway from its center point by ~1/3 m (as much as 1 m) in arbitrary direction during calibration. These assistants also used handheld Bosch 225 ft (68.58 m) Laser Measure devices to determine the device's distance from their reference points (several measurements were taken for each source location), and through a least-squares trilateration procedure [71]

Table 3. Tonal source locations.

| Source Location Identifier | X (m) | Y (m) | Z (m) |
|---|---|---|---|
| P1 | 8.84 | -5.61 | 0.00 |
| P2 | 12.00 | -5.49 | 0.00 |
| P3 | 15.18 | -4.91 | 0.00 |
| P4 | 19.45 | -5.03 | 0.00 |
| P5 | 22.19 | -5.49 | 0.00 |
| P6 | 15.43 | -2.02 | 0.00 |
| P7 | 15.24 | -7.01 | 0.00 |
| P8 | 9.05 | -5.88 | -4.27 |
| P9 | 12.13 | -5.33 | -4.27 |
| P10 | 15.33 | -5.03 | -4.27 |
| P11 | 19.23 | -5.43 | -4.27 |
| P12 | 22.01 | -5.36 | -4.27 |
| P13 | 15.39 | -2.13 | -4.27 |
| P14 | 15.14 | -7.62 | -4.27 |

the device location could always be placed on a Cartesian coordinate system common with the hydrophones.

Tonals were both played and sampled at 192 kHz over two networked MOTU 8M audio interfaces connected by fiber optic to a 2013 Apple desktop running macOS Sierra. Tonals were played using a custom Matlab 2018a script, while sounds were manually saved to the Audacity AUP project format, which accommodated long recording sessions in 16 channels. Multiple versions of Matlab (primarily 2015b, 2018a) were used for downstream data management and handling.

In total, 1,783 of 1,792 sampled tonals were successfully extracted to individual 2-s, 16-channel WAV's (referred to subsequently as "snippets"). This was performed in two steps. First, the raw AUP files were manually separated into 16-channel WAV files containing multiple tonals apiece. Second, a Matlab script read each multiple-tonal WAV, and extracted single-tonal snippets by detecting and orienting to a 0.25-s, 2-kHz tone that was programmed to precede every tonal by 2 s during playing; each snippet window began approximately 1.5 s after the leader tone start. The quality of extraction was confirmed manually. The last preprocessing step at this stage involved band-pass filtering the snippets with a Matlab script based on their maximum and minimum detected tonal frequencies (0.5 kHz gaps were left above and below the maximum and minimum frequencies, respectively) to eliminate a degree of ambient noise.

All snippets were saved with alphanumeric information about their tonal types and source locations for later analysis. Data are available at https://doi.org/10.6084/m9.figshare.7956212. An example of data contained in a single snippet is shown in Fig 4.

## Feature extraction

Overall, our goal was to use information contained in each snippet to perform source localization of the underlying played tonal. In other words, we sought to predict tonal source location from snippet data. We implemented supervised learning techniques to achieve this. This involved computationally building statistical models to predict tonal source location from snippet data by "training" them with a subset of snippet data paired (or labeled) with what we ultimately wished to predict, which included both snippets' discrete source location identifiers (the first column of Table 3) and their source location coordinates (columns two through four of Table 3). A semi-random 90% of snippets were thus labeled and used for model training, and together composed the training data (or set). For the remaining 10% of snippets, which composed the test data (or set), the explicit source location information would remain unknown to the models and used only for evaluating the models' predictions of explicit source location from the snippet data alone. Note that our separation of snippets into training and testing sets was semi-random in that we randomly distributed the snippets under the constraints that the two sets contain a proportional number of snippets from each source location, and that snippets derived from tonals that differed only in their underlying waveform type (sinusoidal or triangular) be distributed together.

To select and cast the snippet data in a way we felt was useful to supervised source localization techniques across a broad range of contexts (including real dolphin whistles manually extracted from recordings), we transformed each snippet into a set of 897,856 numerical features that the models would rely on for source localization (a feature set). In performing this transformation, we needed to comply with our machine learning models' demand that the $i$th feature of every snippet feature set contain information about the snippet's source location that was directly comparable to the information contained in the $i$th feature of every other snippet feature set; in other words, each feature had to measure the same property across all snippets. This last requirement immediately excluded the snippet samples themselves from

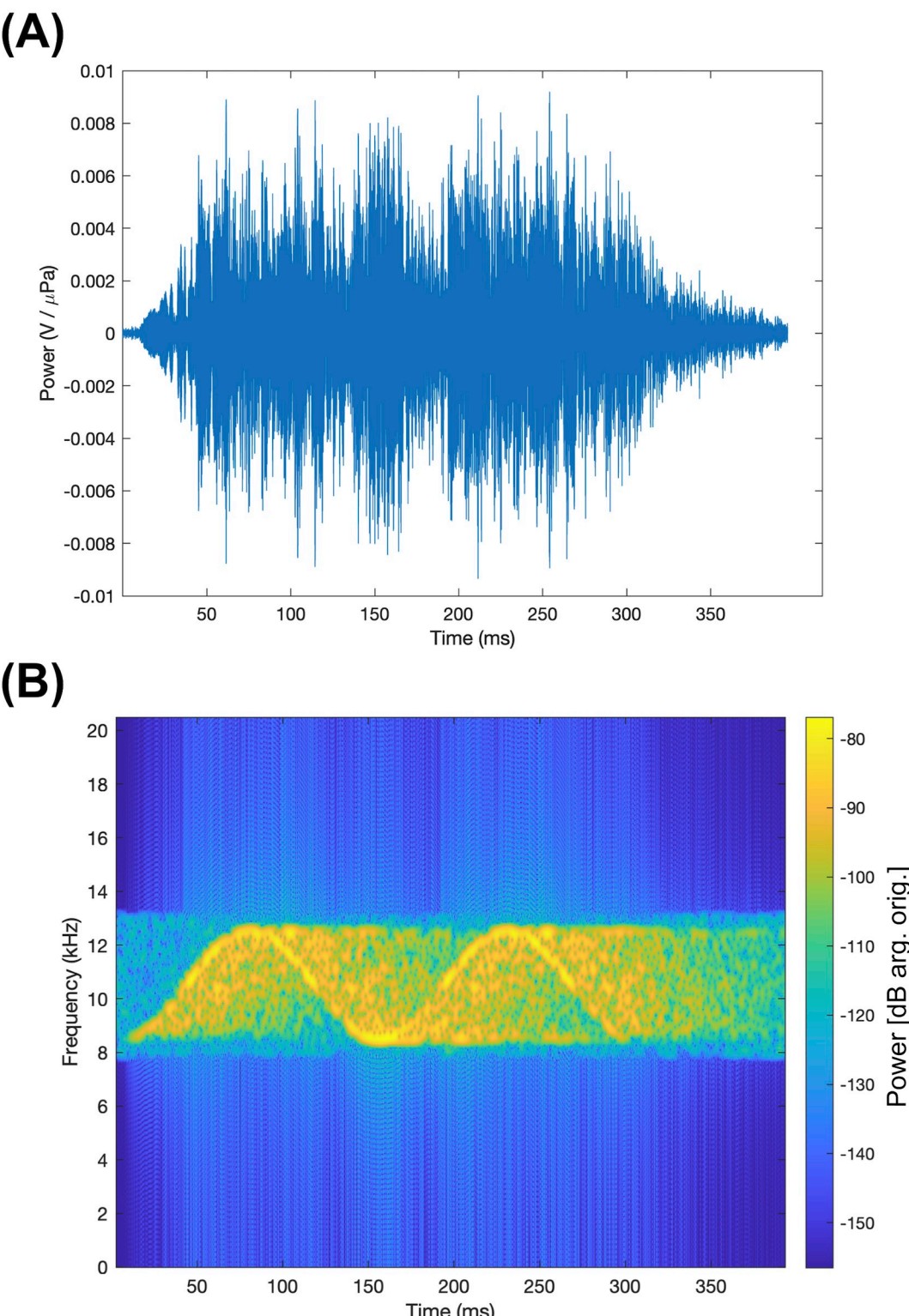

**Fig 4. Example of played signal with reverberation.** Shown here is real sound data obtained by playing the signal from Fig 3 at Source Location P14 and receiving it at Hydrophone 1. (A) The received signal waveform. Signal amplitude is uncalibrated. (B) The spectrogram of the signal waveform. As above, this is a standard, 1024-bin, Hamming-window spectrogram; sampling is 192 kHz, and frequency resolution of the plot is 187.5 Hz (smoothing added). Color scale is in uncalibrated decibels.

consideration as features; for the general applicability of the proposed methods we assumed our tonals were windowed arbitrarily (e.g., that the duration between the snippet start and the tonal "onset" was variable), and that therefore sampling times could not be meaningfully compared across snippets.

Ultimately, we chose features that reflected information about the snippet tonals' time-difference-of-arrivals (TDOA's) to our hydrophones. First, this choice was made because the TDOA's for all pairs of 4 or more hydrophones are theoretically sufficient to determine sound source coordinates using a geometric technique such as Spherical Interpolation, which transform TDOA's into inter-hydrophone distances via the speed of sound and algebraically determine the most probable source-hydrophone distances [31, 33, 67]. Second, this choice was made because a complete set of TDOA's is theoretically available from the information contained in every snippet (which included waveforms from 16 hydrophone channels sharing a clock). Third, this choice allowed us to choose features that were directly comparable across snippets despite variability in their tonal windowing. Lastly, by not choosing features that reflected snippet tonal sound intensity—one possible alternative to TDOA-reflective features—we hoped to produce models that were not limited to classifying sounds of the same source intensity.

Our first features included one estimation of a snippet tonal's TDOA for each unique pair of 16 hydrophones (excluding self-pairs) represented in a snippet, amounting to $\binom{16}{2} = \frac{16!}{2!(16-2)!} = 120$ features that were comparable across snippets. By standard practice, a TDOA between hydrophone $i$ and hydrophone $j$ could be estimated by choosing the relative shift between the two hydrophones' 2 s tonal waveforms that maximized the value of their cross-correlation. We deviated from standard practice only in substituting the standard cross-correlation for the Generalized Cross-Correlation Phase Transform (GCC-PHAT), a form of cross-correlation that equalizes power across frequency bands and allows for alignment of signals based primarily on relative phase shifts [72]. In many contexts, GCC-PHAT has demonstrated better performance than the standard cross-correlation for TDOA estimation [73]. While we found that these TDOA estimations were not reliable enough to accurately perform tonal source localization using Spherical Interpolation, a possibility mentioned in the previous paragraph, we suspected that they still might be useful as part of a larger machine learning feature set.

The next and last 6601 x 136 features consisted of elements from standard, normalized circular cross-correlations [74]: for each unique pair of the 16 hydrophones (including self-pairs) represented in a snippet, 136 in total, we computed the standard, normalized circular cross-correlation for a snippet. We chose to include standard cross-correlation elements because, as discussed last paragraph, they contain information about TDOA's. Similarly, these elements are naturally aligned across snippets regardless of variation in tonal onset time inside a snippet (or other properties such as tonal length); each sample corresponds to a relative time-shift between hydrophone channels that depends only on snippet length and sampling frequency. We normalized the cross-correlations (dividing a snippet's cross-correlation elements by the largest absolute value among them) in order to suppress amplitude information. Our early investigations suggested that these standard cross-correlation elements were more helpful to classification than those derived from the GCC-PHAT, discussed previously. While each correlation series was initially 384,000 elements long (192,000 samples/s x 2 s), we only kept the central 6601 elements from each, corresponding to TDOA information for time shifts up to ~±17 ms between hydrophones. This comfortably ensured we included TDOA information for at least the first arrival of snippet tonals for any pair of hydrophones and any source location (we calculated the longest possible first arrival TDOA to be ~12 ms); without specific

guarantees this also accommodated information about a number of (but certainly not all) TDOA's between later arrivals. Note that later arrival times do not necessarily correspond to greater differences in arrival times (TDOA's). The main purpose of this feature reduction was improved computation times. In general, this reduction is not necessary and may in fact be preferable to avoid.

## Tonal localization by classification, regression and SRP

We began with the task of multiclass classification [75], which entailed training models to predict a tonal's source location identifier from the total set of 14 identifiers (first column of Table 3). The first model class we considered was the Breiman random forest [76, 77]. The random forest class was chosen for several reasons. First, it was chosen for its low susceptibility to erroneously modeling relationships in training/validation data that are not present in the test data, a source of prediction error known as overfitting. Second, the random forest was chosen for its reliable performance across many choices of user-specified input parameters, also called hyperparameter values. Lastly, it was chosen for its ability to provide ancillary information about the input features, specifically information helpful to feature set reduction; this is the common goal of intelligently merging/eliminating features in the input data without loss of prediction accuracy, for increased computational performance and possibly increased accuracy. This ancillary information consisted of a measure of each feature's relative importance to classification. The specific measure we derived from the random forest is termed the permuted variable delta error, referring to the increase in classification error when a given feature is effectively randomized (representing an estimation of that feature's importance).

Except where indicated, we built the random forest using Matlab 2018a's "TreeBagger" class with default hyperparameter values. This allowed us to grow a Breiman random forest composed of classification trees (built from the CART, Classification And Regression Tree, algorithm [78]) on the training data; each tree was sequentially trained on a random ~75% subset of the training set snippets (labeled with their corresponding source location identifiers), using a random $\sim\sqrt{897,856}$-element subset of the total available features (as per standard practice). Out-of-bag (OOB) error, referring to the classification error on snippets not randomly chosen for the training subset, was used to evaluate validation accuracy. Validation accuracy (or error) is a pre-testing measure of model performance that is often repeatedly calculated while tuning a model's hyperparameters (which was not done here). Note that background on the methods we considered are available at MathWorks (https://www.mathworks.com) and scikit-learn (https://scikit-learn.org/stable/).

Finally, we evaluated the random forest's accuracy in predicting the source location identifiers of the test snippets, which were fed to the classifier without source location identifier labels. As for all classification models, we measured the random forest's classification accuracy as the simple percentage of source location identifiers correctly predicted from the test set snippets. Where-ever classification accuracy is accompanied by a confidence interval, this is 95% confidence interval was computed as the Wilson confidence interval, which is a function of the sample size.

Next, we used the permuted variable delta error as a measure of feature importance to obtain a reduced feature set ahead of training models from additional classification and regression classes. Our reduced feature set included all features that possessed nonzero permuted variable delta error (ultimately, the random forest did not use most features for classification). Using this new set of features, we similarly trained models from classes including the basic CART classification tree [75, 78], the linear and quadratic Support Vector Machine (SVM) [75, 79], and linear discriminant analysis [80]. Again, these standard multiclass classification

models were built with Matlab 2018a, using the "fitctree," "fitcecoc," and "fitcdiscr" classes (respectively) with default hyperparameter values except where indicated. In particular, based on evaluation on the training/validation set we limited the number of the classification tree's "splits" (effectively, a measure of the complexity of the tree's decision making) to 20, to reduce this model class' natural tendency to overfit. These model classes were initially evaluated with 10-fold cross validation on the training/validation data set (after randomly splitting the training/validation data into 10 groups, the accuracy of a model trained on 9 groups was evaluated on the 10th group, the result averaged across testing on all 10 groups), before the training data were combined to train models for evaluation on the test data set.

After considering tonal source location prediction by classification, we next considered source location prediction by regression. Whereas the classification models were designed to predict a snippet tonal's source location from the set of 14 source location identifiers, the regression models were designed to predict a snippet tonal's three-dimensional source coordinates from anywhere in three-dimensional space. Thus, training snippets were now labeled with source location coordinates rather than source location identifiers.

We specifically considered Gaussian process regression (also termed *kriging*) [81, 82]. This form of regression performs predictive interpolation by merging the weighted labels of nearby training points, with weights empirically determined by calculating autocorrelations between points (i.e., their tendency towards similarity) and inferring trends in the underlying feature landscape. The method was developed in geostatistics and is correspondingly suited to interpolation over smoothly varying landscapes, and where appropriate its assumptions are powerful for interpolating within data that is sparely sampled, as our data from 14 source locations were. Judging by validation error on our training/validation data, we determined Gaussian regression was more effective than ordinary regression methods on our data; we used a squared exponential kernel function and otherwise the default hyperparameters belonging go the "fitrgp" class in Matlab 2019b. Gaussian regression models were built and evaluated in two ways. In the first way, the reduced feature sets of all training snippets were used to build three models for predicting snippet tonal source coordinates; one model was built to predict each of the three source coordinates. These three models were then evaluated on all test snippets. In the second way, the reduced feature sets of training snippets from all but one of the fourteen source locations were likewise used to build three models. These models were then evaluated only on test snippets from the excluded source location; in other words, the models were prompted to predict the source coordinates of snippet tonals from novel locations for which they received no training, to perform spatial interpolation. Three regression models were built and evaluated with every source location excluded from training and used for evaluation in turn, and the results aggregated. While we were doubtful of our models' ability to precisely interpolate at distances of several meters from 14 points, this second way of building and evaluating the models was envisioned to show that reduced-feature-set Gaussian regression models are capable of a degree of spatial interpolation on our problem.

For evaluation of our regression models, we implemented a Steered-Response Power approach to localizing the snippet tonals in the test set, using established methodology involving spatial gridding [59] together with what details were made available by Thomas et al. [60], primarily pertaining to limiting the search field. We gridded the pool into 15 cm cubes, with cube corners representing potential tonal source locations. For a hypothetical tonal from each corner, we calculated the full set of expected TDOA's for our 16 hydrophones (as before, 120 TDOA's). We then shifted the 16 waveforms of a snippet in our test set by each set of TDOA's, simulating the reversal of the relative shifts caused by travel from each grid corner, before summing the dot products of every unique waveform pair. In theory, using the TDOA set from a snippet tonal's true source location would maximize this final quantity. Consequently, the

source location predicted by SRP corresponded to the cube corner whose TDOA's produced the largest value. Unlike the sound source localization methods we propose, SRP required the speed of sound as input, which we calculated to be ~1533 m/s from the Del Grosso equation [83]; the pool salinity was 31.5 ppt and temperature 26.04˚C. As an aside, we attempted to replace the standard cross-correlation in the prior calculation with the Generalized Cross Correlation with Phase Transform [72], which constitutes the SRP-PHAT technique for sound localization discussed in [59], but found the results to be inferior.

We first compared the performance of Gaussian process regression with SRP by comparing their source prediction/localization errors, in particular the deviations (simple differences) between model-estimated source coordinates and the physically measured source coordinates. We calculated both Euclidean (alternatively, straight-line or 3D) deviations and deviations along the three individual component axes, using the Median Absolute Deviation (MAD) metric bounded by standard 25th-75th percentile Interquartile Ranges (IQR) to combine and express this information across all test set snippets; one MAD and set of IQR's was computed for each model class along each axis of interest. We also generated histograms of the absolute deviations, to illustrate their distributions for different model-axis sets. While the MAD and IQR metrics allowed us to compare model performances in a way that was intuitive, strictly speaking they were not appropriate for determining whether the MAD's were statistically distinct. To make such comparisons given that the sets of absolute deviations were not necessarily drawn from normal distributions (which we established with Anderson-Darling tests), we first used a Krusal-Wallis test (a nonparametric equivalent of the ANOVA) to show that the different sets of absolute deviations likely did not all originate from the same distribution. Second, we used Dunn's post-hoc tests (akin to Tukey's tests used after an ANOVA) to determine whether individual pairs of MAD's significantly differed at a 5% significance level, which was determined from a comparison of the nonparametric "mean ranks" values.

Finally, we sought confirmation that our features were used by the machine learning models to distinguish source locations based on inferred TDOA information and geometric reasoning. Noting that data from two hydrophones in a single 4-hydrophone-array would not as helpful to TDOA-based, geometric sound source localization as two hydrophones in two different 4-hydrophone-arrays (owing to larger changes in their TDOA's with changes in source location), we predicted that direct or indirect information from all possible pairs of the four 4-hydrophone-arrays would be necessary for optimum classification. We evaluated our prediction by looking more closely at the classification tree previously constructed, which we found to be naturally "shallow" (composed of few branches) and parsimonious, selecting a minimum (or near-minimum) subset of features for use in classification. We then mapped these features' importance (again, using the permuted variable delta error from a random forest, constructed as before but on the new feature set) back to the hydrophone and 4-hydrophone-array pairs from which they were derived, as a measure of which hydrophone and 4-hydrophone-array pairs were most important to classification.

## Results

The random forest classification model trained on the full feature set reached 100.0% OOB accuracy at a size of ~180 trees. We continued training to 300 trees, and evaluated the resulting model on the test set: 100.0% accuracy was achieved, with 6,778 features possessing permuted variable delta error greater than 0 (based on OOB evaluations). Note that, because random forest construction did not consider all features equally (even excluding some) given that each member tree used a random subset of available features during training, subsets of features other than this set of 6,778 could potentially accommodate 100.0% test accuracy. When we

**Table 4. Accuracy of source location prediction by classification.**

| Model | 10-fold CV Accuracy (%) | Test Accuracy [Wilson CI] (%) |
|---|---|---|
| Classification Tree | 96.90 | 97.75 [97.06–98.44] |
| Linear SVM | 100.0 | 99.44 [98.34–100.0] |
| Quadratic SVM | 100.0 | 100.0 [n/a] |
| LDA | 100.0 | 100.0 [n/a] |

considered which 4-hydrophone-array pairs the 6,778 TDOA and cross-correlation features represented, we found that all pairs of the 4-hydrophone-arrays were represented with no significant preference.

We trained several more models on the reduced, 6,778-element feature set, including a basic classification tree, a linear and quadratic SVM, and linear discriminant analysis (LDA), performing initial evaluations with 10-fold cross-validation before aggregating all training data for final evaluation on all test data. Results are shown in Table 4.

Using the reduced feature set, we performed Gaussian process regression (kriging) to predict source coordinates of test snippet tonals, training one model for each Cartesian axis. We trained and evaluated models in two ways. First, we trained three models on all training/validation data for evaluation on all test data; this method is referred to as Standard Gaussian Regression (S GR) below. A random subset of predictions made this way is plotted in Fig 5. Second, we trained three models on training/validation data from all but one source location for evaluation on test data exclusively from the excluded source location, repeating the process

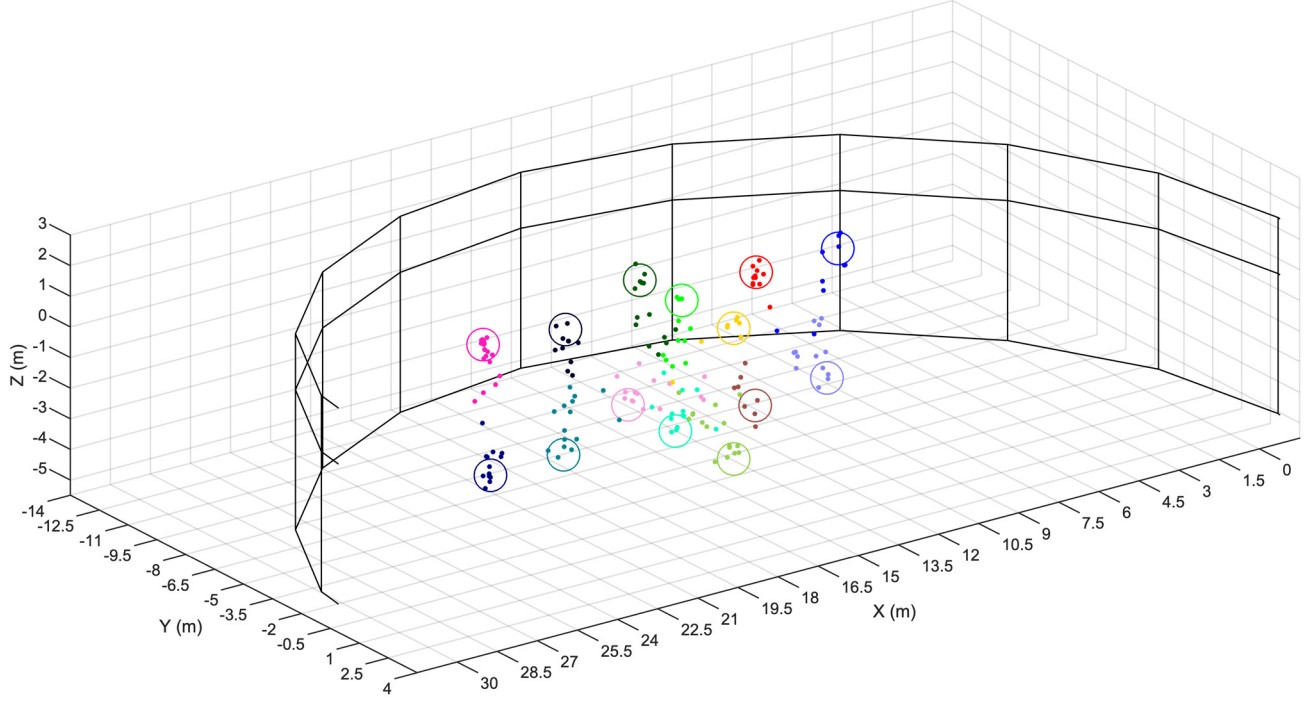

**Fig 5. Predictions of test set source coordinates by Gaussian process regression models.** The half-cylindrical National Aquarium EP is depicted. Large unfilled circles indicate the true source coordinates of the test snippet tonals; each has a unique color. Small filled circles indicate the source coordinates of test snippet tonals predicted by Gaussian process regression, colors matching their respective true coordinates.

**Table 5. MAD and IQR of test set source coordinate predictions.**

| Model | Axis | MAD [IQR] (m) |
|---|---|---|
| S GR | Eucl. | 0.66 [0.34–1.57] |
| S GR | X | 0.19 [0.07–0.39] |
| S GR | Y | 0.13 [0.05–0.34] |
| S GR | Z | 0.52 [0.16–1.18] |
| LO GR | Eucl. | 3.37 [2.85–3.75] |
| LO GR | X | 0.56 [0.26–1.03] |
| LO GR | Y | 0.50 [0.17–1.62] |
| LO GR | Z | 2.73 [2.14–3.39] |
| SRP | Eucl. | 1.56 [0.73–2.48] |
| SRP | X | 0.43 [0.18–0.73] |
| SRP | Y | 0.52 [0.21–0.98] |
| SRP | Z | 0.91 [0.15–1.98] |

for all permutations of source locations and aggregating the predictions; this method is referred to as Leave-Out Gaussian Regression (LO GR) below. For comparison, we also predicted test set source coordinates using SRP. For all sets of predictions, we computed absolute deviations, which are the basis of the statistics in Table 5, Figs 6 and 7. Our use of nonparametric statistics was motivated by Anderson-Darling tests that rejected the null hypotheses that the sets of absolute deviations were drawn from normal distributions (5% significance level). Further, a Kruskal-Wallis test on the deviation sets' mean ranks determined the sets did not originate from the same distribution at a 5% significance level.

The classification tree previously built on the reduced, 6,788-element feature set displayed the unique property of parsimony, in that it naturally identified a smaller, 22-element feature subset as sufficient for performing source location classification. We trained a parsimonious random forest on this 22-element feature set (using the same hyperparameter values as before), this model achieving 98.88% test accuracy (95% CI [97.73—100.0]). Thus, we considered this 22-element feature set both sufficient and small enough to determine whether classification made use of features derived from a mixed set of spatially distant hydrophone and 4-hydrophone-array pairs, as would be favored by a geometric, TDOA-based approach to sound source localization (such as Spherical Interpolation). The permuted variable delta error was summed across hydrophone and then averaged across 4-hydrophone-array pairs, which is visualized in Fig 8. Overall, we note that features representing all pairs of 4-hydrophone-arrays, directly or through a transitive relationship (e.g., in Fig 8 Panel D, TDOA information between arrays 3 and 4 is implied by TDOA information between arrays 1 and 3 and between arrays 1 and 4), are utilized for classification.

## Discussion

We provide a basic proof of concept that sound source localization of bottlenose dolphin whistles might be achieved with classification and regression methods in a half-cylindrical captive dolphin enclosure, by localizing the semi-stationary sources of artificial tonal sounds. An enclosure of this type traditionally poses challenges to source localization of tonal sounds (by tag-less methods) given the prominence of multipath effects, which complicate the acquisition of TDOA's by cross-correlation methods for successful use with standard geometric sound source localization methods (such as Spherical Interpolation). Moreover, for the same conditions we showed that, for the localization of test set tonals played near training set tonals

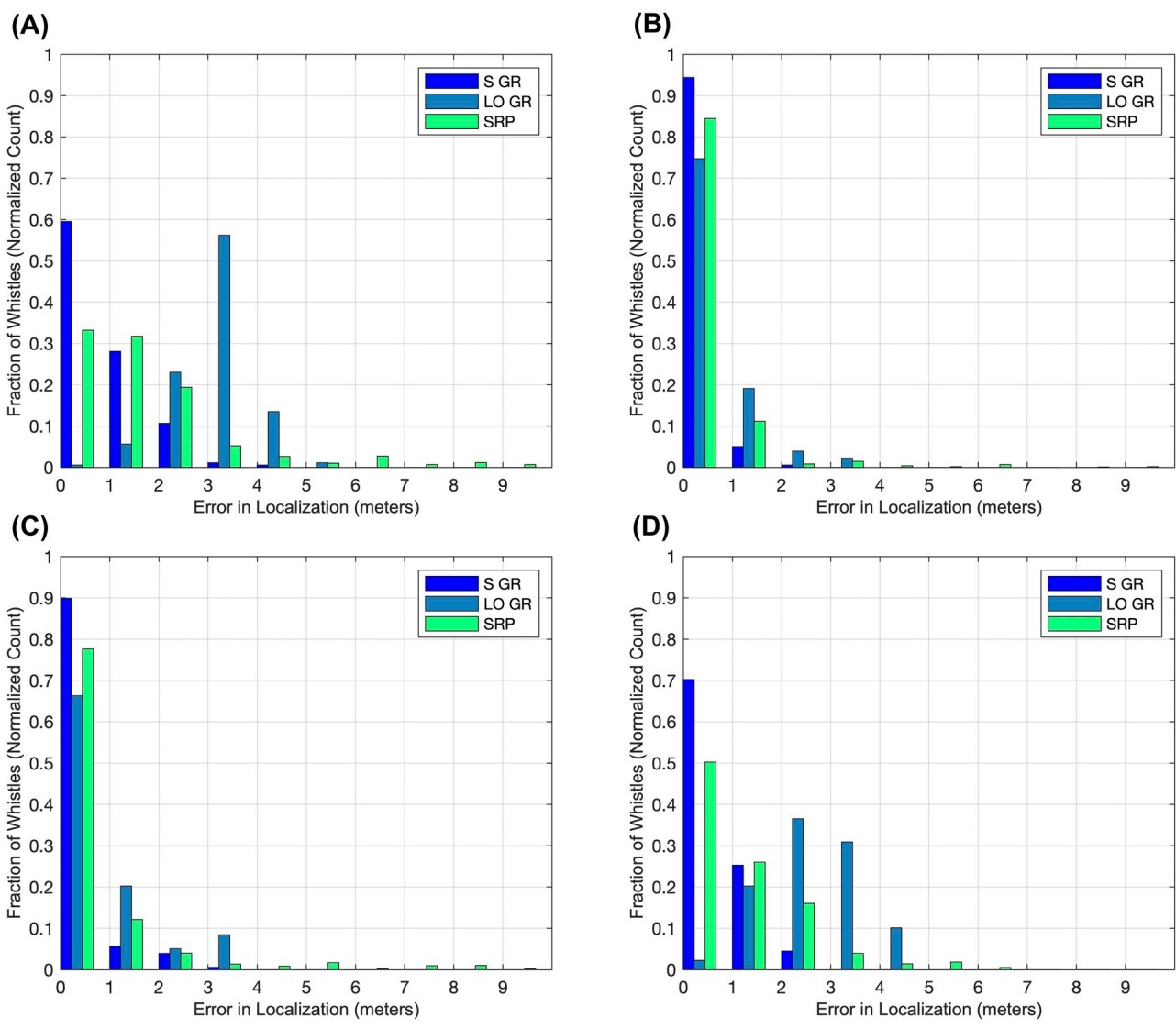

**Fig 6. Multi bar graphs displaying histogram data for the test set source coordinate prediction error sets (3 methods x 4 axes).** For the Standard Gaussian Regression (S GR), Leave-Out Gaussian Regression (LO GR), and SRP methods, test snippet tonal localization error (absolute deviation) is displayed (A) for the Euclidean (straight-line) direction, (B) for the X-axis, (C) for the Y-axis, and (D) for the Z-axis.

(within ~1/3 m, the speaker sway range), Gaussian regression outperforms an established Steered-Response Power (SRP) method for whistle source localization. Localizing in the plane of the pool surface, Gaussian process models localized tonals played at source locations not included in training with similar accuracy to SRP, interpolating at distances as long as several meters. This is significant because source localization along the pool surface can be sufficient for attributing whistles to dolphins, as dolphins often occupy distinct locations in this plane. Moreover, this is often the only plane for which imaging is available (i.e., using an overhead camera) and in which the dolphins can be assigned position coordinates, and thus potentially the only plane in which sound source coordinates are useful to whistle source attribution.

The initial data consisted of 2-s, 16-channel WAV samples ("snippets") of 127 unique bottlenose-whistle-like tonals played at 14 source locations inside the National Aquarium EP. Each of the 16 channels corresponded to one hydrophone among four 4-hydrophone-arrays.

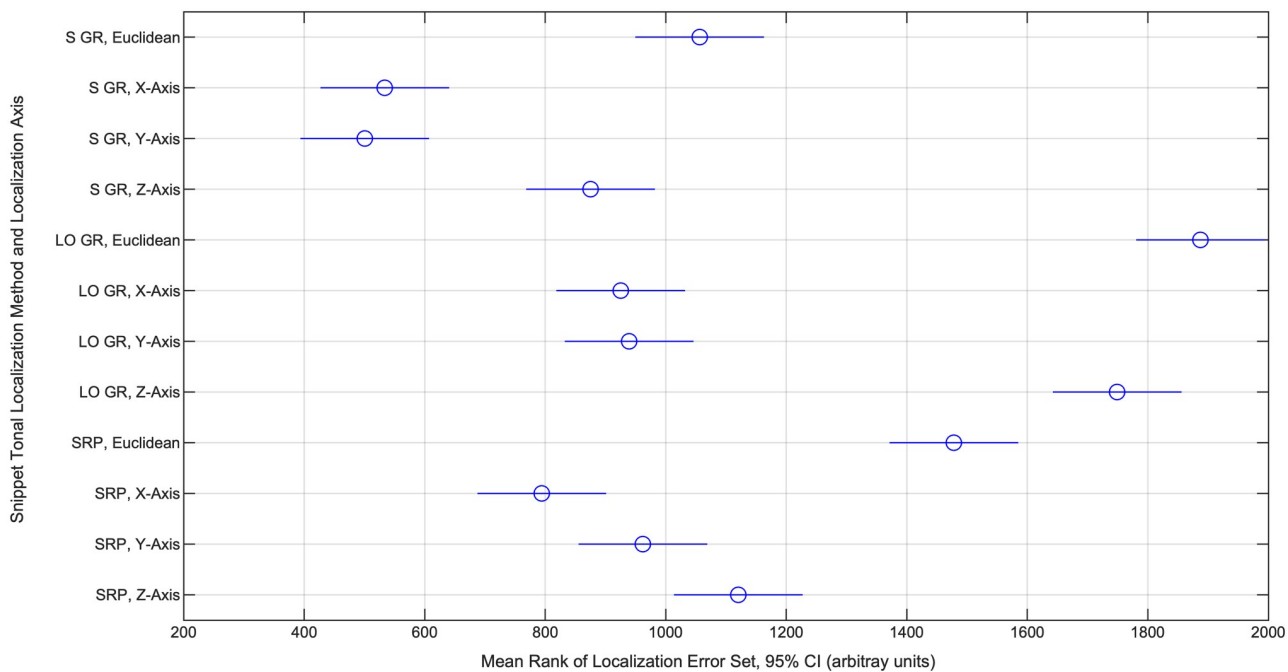

**Fig 7. Mean ranks of test set source coordinate prediction error sets (3 methods x 4 axes).** The mean ranks for the localization error (absolute deviation) sets belonging to every source localization method and axis of localization, computed ahead of a Kruskal-Wallis test, are shown with 95% CI's. Based on Dunn's tests with an overall 5% significance threshold, overlapping intervals reflect sets whose underlying distributions are not statistically distinct (implying non-distinct MAD's); non-overlapping intervals reflect sets that are distinct (implying distinct MAD's).

The 1,783 good-quality snippets were semi-randomly divided into training/validation and test sets in a 9:1 ratio for model building/training and evaluation, respectively.

First, we showed that a random forest classifier with fewer than 200 trees achieved 100% testing accuracy at predicting from which of the 14 source locations a snippet tonal originated. The classifier used 6,778 of 897,856 available features (those features with nonzero permuted variable delta error, or equivalently nonzero feature importance), which included TDOA's obtained from GCC-PHAT as well as normalized cross-correlations from all pairs of hydrophones. We then showed that linear discriminant analysis and a quadratic SVM achieved the same classification accuracy on the same reduced, 6,778-element feature set. Despite the classification models' success, we concede that the distant spacing between adjacent source locations in our data are unlikely to be conducive to the practical localization of dolphin whistles for source attribution. However, were these model classes to achieve similar accuracy when trained on data reflecting shorter spacing between adjacent source locations (spacings of approximately 0.5—1.0 m, noting some spacings in our present data were less than 2 m), the linear model classes might offer simple and computationally efficient source localization of dolphin whistles for attribution.

Although it remains unclear to what extent test snippet tonals corresponding to novel source locations not included in the training set are classified to their most logical (i.e., nearest) training set source locations, we note that our classifiers' success was achieved despite the ~1/3 m drift of the speaker during play-time. This together with the success of regression may indicate a degree of smoothness in the classifiers' decision-making, and the likelihood of classification of snippet tonals from novel locations to nearby training set source locations. Also, it is reassuring that a linear classifier, which by definition cannot support nonlinear decision making, can achieve high accuracy: because we expect the TDOA and cross-correlation features to

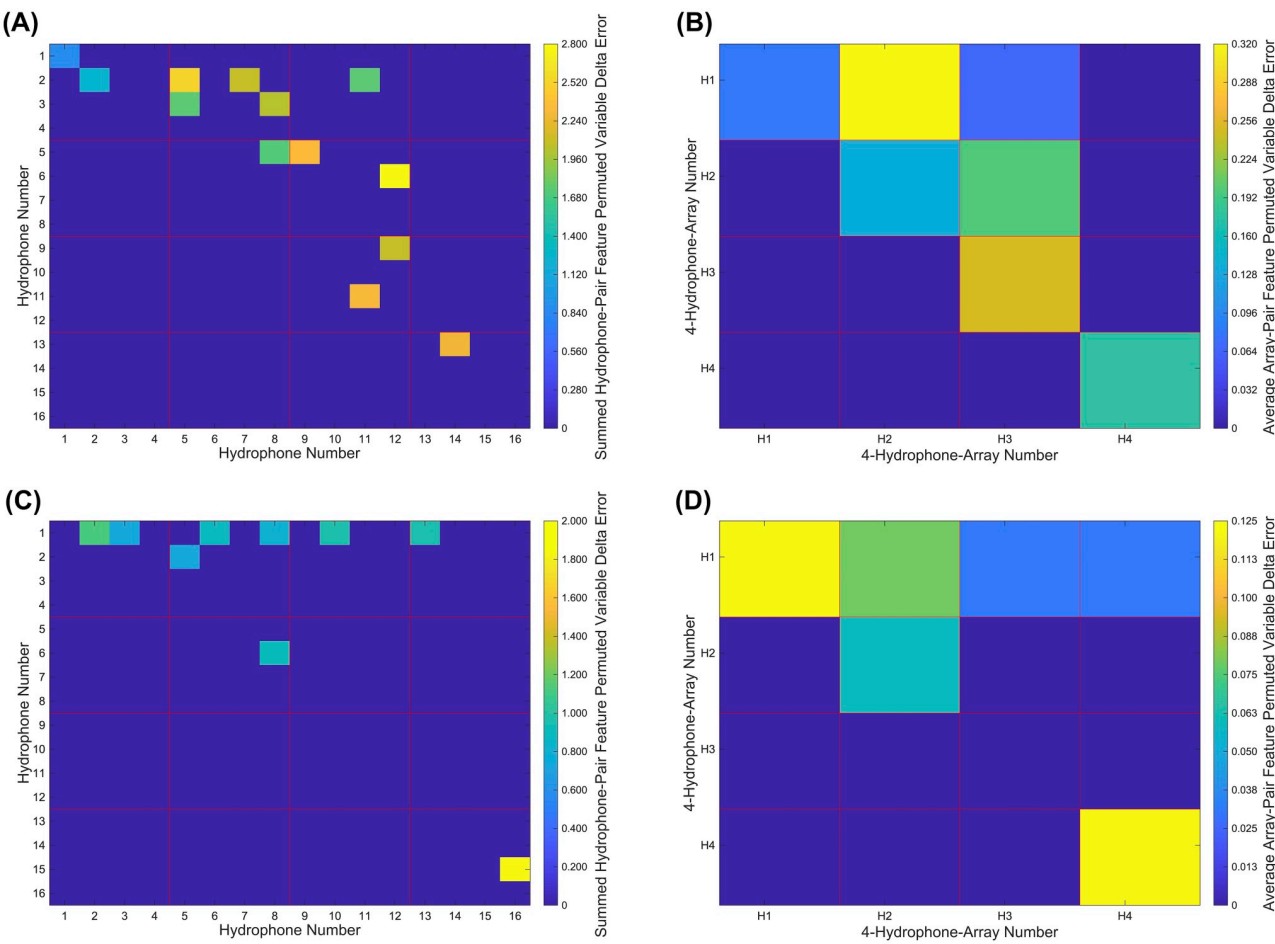

**Fig 8. Cross-hydrophone and cross-4-hydrophone-array feature importances for the parsimonious random forest.** Feature importance values for a parsimonious, 22-feature random forest model summed within corresponding hydrophone pairs to make (A) and (C), with those values subsequently averaged within 4-hydrophone-arrays to make (B) and (D). Note that values are equal between reversed hydrophone/4-hydrophone-array pairs (e.g., values are equal for hydrophone pairs 1-2 and 2-1) but plotted only once. (A) Cross-hydrophone importances for cross-correlation features. (B) Cross-4-hydrophone-array importances for cross-correlation features. (C) Cross-hydrophone importances for TDOA features. (D) Cross-4-hydrophone-array importances for TDOA features.

vary continuously over real space, the (linear) hyperplanes that partition classification zones in feature space—the mathematical basis of linear classification—suggest continuous classification zones in real space. Nevertheless, the nature of the models' classification of snippet tonals from novel source locations to training set source locations still warrants further investigation.

We more suitably addressed the localization of test snippet tonals from source locations that were novel to the models by using Gaussian process regression, which we used to predict the source coordinates of test snippet tonals (with one regression model trained to each of the three coordinates). This model class was rather successful when trained on the full training data, achieving Euclidean test MAD of 0.66 m (IQR = 0.34—1.57). In order to better assess the regression models' capacity for long-distance interpolation, we evaluated the regression models' performance on test snippets for which no training snippets from the same source locations were used for model training; in effect, the models were prompted to predict the source coordinates of test snippet tonals from novel locations. While the regression models' overall performance on test snippets representing novel source locations was not satisfactory,

admitting error larger than average dolphin length (MAD of 3.37 m, IQR = 2.85—3.75), when we decomposed the error along three Cartesian axes (X-axis MAD of 0.56 m with IQR = 0.26—1.03, Y-axis MAD of 0.50 m with IQR = 0.17—1.62, and Z-axis MAD of 2.73 m with IQR = 2.14—3.39), we found that the Euclidean localization error was dominated by localization error along the Z-Axis, or direction of pool depth (Fig 7). This is significant because only two pool distinct pool depths were represented in our data, which are intuitively insufficient for meaningful interpolation in this direction. We think it reasonable to suggest that interpolation along the Z-axis, and thereby overall Euclidean interpolation, would substantially improve with finer depth-wise spatial sampling among the training data. Increasing spatial sampling from our 14 points in the lateral directions might be expected to improve Euclidean interpolation as well. Nevertheless, we note that MAD for the interpolative case in the lateral directions was still less than average adult dolphin body length, if still greater than the minimum possible separation between two dolphins' whistle-transmitting heads (perhaps 1/3 m).

To benchmark the Gaussian process regression models, we also predicted test snippet tonal source coordinates using a standard SRP approach that has met success elsewhere [60]. Gaussian regression significantly outperformed SRP when all training data were used (Fig 7). When the regression models were deprived of training data from source locations of evaluated test snippets (the novel source location or long-distance interpolative case), SRP performed better overall and along the Z-axis; there was no significant performance difference along the other two component axes. While this indicates that, with the current training data, Gaussian regression does not outperform SRP in three dimensions when prompted to interpolate at longer distances, it also indicates that Gaussian regression is capable of long-distance interpolation with comparable accuracy as SRP in the lateral directions. Further, related to the previous discussion, we might expect Gaussian process regression to perform at least as well as SRP were more than two pool depths represented in the training data. Most importantly, the results suggest that Gaussian process regression can perform just as well as SRP for localizing tonals across the pool surface (i.e., disregarding depth), which is often sufficient for distinguishing among potential sound sources based on overhead imaging (as in Thomas et al.). Thus, even with our limited training data, Gaussian process regression seems to be a promising option for whistle localization ahead of whistle attribution in an environment with pronounced multipath effects.

Lastly, we showed that an extremely sparse, 22-feature random forest with high classification accuracy (98.88%)—a parsimonious model—includes direct or indirect TDOA information from all possible pairs of the four 4-hydrophone-arrays. The information, manifested in the 22 features that included TDOA estimates from GCC-PHAT and normalized cross-correlation elements for pairs of hydrophone channels, reflected the largest of the temporal separations among pairs of our 16 hydrophones. As increasing the temporal separation of hydrophones is expected to optimize the performance of TDOA-based geometric methods of sound source localization (such as Spherical Interpolation), which in fact motivated our initial placement of the four 4-hydrophone-arrays, the parsimonious classifiers' selection of these features suggests that they are reproducing similar geometric reasoning in their decision making. However, we concede that the inner logic of the models ultimately remains unknown.

## Caveats

These methods must be still be evaluated on tonals originating from dolphins (i.e., whistles). Not only do dolphins move faster than our speaker, but they possess different acoustical directional properties, and their whistles potentially fall outside of the "whistle space" spanned by

our tonals. Regarding movement speed, we are hopeful because the past success of the Thomas et al. SRP method on dolphin whistles suggests that whistle cross-correlations, which both methods rely on, retain information sufficient for localization even when a source moves at dolphin speed. Regarding whistle directionality, though the directionality of whistles strictly differs from that of our speaker-produced tonals, it has been observed that the "fundamental" or lowest-frequency trace of whistles (the focus of our method) is approximately omnidirectional [84]. Lastly, regarding whistles existing outside of our artificial "whistle space," we note that dolphin groups of the approximate size of the National Aquarium's seven have been shown to possess fewer than 100 unique call types [21]. This suggests the possibility of creating a complete, customized set of training sounds for any small group of dolphins. A final consideration is whether these methods are applicable to tonals/whistles originating simultaneously from more than one source; while these cases have been rare at the National Aquarium, they would likely factor into the methods' utility in the wild. This case must be still be studied, but we would expect our methods to succeed (dependent on the previous considerations) in cases where whistle fundamentals can be cleanly separated by signal preprocessing.

## Conclusions and outlook

We feel this study offers a valid argument that machine learning methods are promising for solving the problem of bottlenose whistle localization in highly reverberant aquaria, where tag-based solutions to whistle source attribution are not feasible. We offer evidence to suggest that these methods might be capable of greater accuracy than SRP methods given adsquate training data, coming at smaller real-time computational expense—at the cost of initial model training. While we caution that these methods still must be trained and evaluated on whistles originating from real dolphins, we opine that our results are rather encouraging.

And looking further, there are a huge number of scientific areas where researchers need to localize the origin coordinates of sounds of interest to them. Because both the nature of the sounds and the nature of the acoustical environment matter to how this problem is solved, there's a massive spread of methods tailored to the specific sounds and the specific acoustical environment. Yet one problem that often recurs in many situations is that of reverberations and multiple closely-spaced echoes ("multipath") causing confounds that badly impact localization. While we are motivated specifically by the problem of tracking dolphin vocalizations in a man-made aquarium pool, we believe that the solution we have found in our specific problem has great potential for broad applicability. Our solution is to harness the power of regularized machine learning to, *de facto*, map out and learn the idiosyncrasies of the acoustical environment; once the environment has been learned, localization can be both accurate and computationally fast even in the face of massive reverberation.

## Acknowledgments

We thank the many helpers, support staff, and backers involved this project. At the Rockefeller University, we thank Ana Hočevar, Sanjee Abeytunge, Vadim Sherman, Brigid Maloney, Dimitrios Moirogiannis, A. James Hudspeth, Alipasha Vaziri, and Fernando Nottebohm. At Hunter College, we thank Megan McGrath, Stephanie Bousseau, Raymond Van Steyn, Miranda Trapani, Jennifer Savoie, Eric Ramos, Kristi Collom, Adrienne Koepke, Robert Dutchen, and Ofer Tchernichovski. At the National Aquarium, we thank James Tunney, Manuel Chico, Richard Snader, Allan Consolati, Kerry Diehl, Susie Rodenkirchen Walker, Allison Ginsburg, April Martin, Kimmy Barron, Rebekah Miller, Kelsey Fairhurst, Gretchen Geiger, Kelsey Wood, Angela Lopresti, Nicole Guyton, Leigh Clayton, Brent Whitaker, Jill

Arnold, Mark Kennedy, Holly Bourbon, Andrew Pulver, and John Racanelli. We thank unaffiliated helper Elias Buchanan Ohrstrom.

## Author Contributions

**Conceptualization:** Sean F. Woodward.

**Data curation:** Sean F. Woodward.

**Formal analysis:** Sean F. Woodward.

**Funding acquisition:** Diana Reiss, Marcelo O. Magnasco.

**Investigation:** Sean F. Woodward.

**Methodology:** Sean F. Woodward.

**Resources:** Diana Reiss, Marcelo O. Magnasco.

**Software:** Sean F. Woodward.

**Supervision:** Diana Reiss, Marcelo O. Magnasco.

**Validation:** Sean F. Woodward, Diana Reiss, Marcelo O. Magnasco.

**Visualization:** Sean F. Woodward.

**Writing – original draft:** Sean F. Woodward.

**Writing – review & editing:** Sean F. Woodward, Diana Reiss, Marcelo O. Magnasco.

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
