## [Decision Letter · Decision Letter 0]

9 Jan 2020

PONE-D-19-33956

Machine Source Localization of *Tursiops truncatus* Whistle-like Sounds in a Reverberant Aquatic Environment

PLOS ONE

Dear Dr. Woodward,

Thank you for submitting your manuscript to PLOS ONE. After careful consideration, we feel that it has merit but does not fully meet PLOS ONE’s publication criteria as it currently stands. Therefore, we invite you to submit a revised version of the manuscript that addresses the points raised during the review process.

We would appreciate receiving your revised manuscript by Feb 23 2020 11:59PM. To enhance the reproducibility of your results, we recommend that if applicable you deposit your laboratory protocols in protocols.io, where a protocol can be assigned its own identifier (DOI) such that it can be cited independently in the future. For instructions see: http://journals.plos.org/plosone/s/submission-guidelines#loc-laboratory-protocols

We look forward to receiving your revised manuscript.

Kind regards,

Haru Matsumoto

Academic Editor

PLOS ONE

Journal Requirements:

'We thank the National Aquarium for participating in this study, as well the National 402

Science Foundation (Awards 1530544, 1607280), the Eric and Wendy Schmidt Fund for 403

Strategic Innovation, and the Rockefeller University for funding.'

'MO Magnasco, DR Reiss

Awards 530544, 1607280

National Science Foundation

https://www.nsf.gov

The funders had no role in study design, data collection and analysis, decision to

publish, or preparation of the manuscript.'

Additional Editor Comments (if provided):

Dear Dr. Woodward,

The manuscript has improved but it is still lacking details and scientific discussions. Although the application of machine learning to animal localization is unique and the results are interesting, I have to agree with both reviewers that the manuscript needs a major revision. As guidelines, 1) as reviewer 2 pointed out, PLOS One draws a broader audience and not just for experts in machine learning or bio-acoustics. For that, it needs more explanation without losing your audience. 2) The experiment must be repeatable with details of experimental set-up (e.g., hydrophone locations in XYZ as pointed out by both reviewers). I am sure that marine bio-acoustics researchers are eager to apply your ML method to the other animals in the ocean if you can describe the experiment clearly with a few jargon. Unlike the ocean, the aquarium tank is a controlled environment. 3) Re-submission without the details of your aquarium experiment set up and methodology, especially the dimensional information, will jeopardize further review. 4) Before resubmitting the revision, please ask your coauthors to proofread your manuscript carefully in order to save the reviewer's time.

Regards,

Haru Matsumoto

Reviewers' comments:

Reviewer's Responses to Questions

**Comments to the Author**

1. Is the manuscript technically sound, and do the data support the conclusions?

Reviewer #1: Partly

Reviewer #2: Partly

2. Has the statistical analysis been performed appropriately and rigorously? 

Reviewer #1: No

Reviewer #2: Yes

3. Have the authors made all data underlying the findings in their manuscript fully available?

Reviewer #1: Yes

Reviewer #2: Yes

4. Is the manuscript presented in an intelligible fashion and written in standard English?

Reviewer #1: No

Reviewer #2: No

5. Review Comments to the Author

Reviewer #1: This paper aims to tackle an important challenge of localization: associate the recorded whistle to the sound-producing animal. Instead of TDOA-based localization algorithms, the authors used classify the approximate locations and estimate the locations (regression) through random forest, support vector machine and other machine learning algorithms. It was a promising work but the data could be summarized more clearly, experiment setup laid out more precisely and results reported more systematically. It seems to be a work written by a junior academic member without proof-reading from other co-authors. There're simply too many errors and inconsistency all over the manuscript. I recognized that there might be a diamond in the rough. However, the paper have to be organized in a better shape so that scientific findings can be communicated effectively.

Detailed comments are as follows:

In abstract, please refrain from using abbreviations without the full names. What is MAD? What is IQR? Why is SRP the abbreviation of Steered-Response? What does P stand for?

It's not clear what it means by "...comparable accuracy - even when interpolating at several meters - in the lateral directions when deprived of training data at testing sites..." and "...in all domains..."

Line 120-123: It'll be easier to understand through an equation or an example on the figure.

Line 127: It's almost impossible to imagine what y(t) might look like. The denominator arcsin(Phi(t)) are very likely to be incorrect since the input to arcsin needs to be [-1, 1] whereas Phi(t) can be outside this range.

Line 129: Why does m=0.8 correspond to two harmonics?

Line 133: What is "pre-speaker"?

In the legend of Figure 1, it says duration = 1 s whereas Figure 1 only shows roughly 340 ms. In addition, please use SI units everywhere in the manuscript, i.e., "s" instead of "second" or "sec". Details and examples of SI units can be seen:

https://www.bipm.org/en/measurement-units/

Line 151: what are the depths of the hydrophone? What's the separation of the four hydrophones in a location? If there's no significant separation of depth, how can you expect that they are able to locate the depth of locations?

Line 163: the authors mentioned "Audacity AUP sound format". AUP is not sound format. It's Audacity's format for organizing projects.

Line 175: "...with sinusoids and quasi-sinusoids with the same parameters grouped together given their close similarity." There might be too many "with"s.

Line 205 - 207: Why does 27,126 Fourier transform elements correspond to 0 to 27,126 Hz? What is your FFT/DFT size? In order to have 1 s frequency resolution, DFT size needs to be 192,000!

Line 210: Why is the number of features 897,871? I calculated it to be 897,736 (=6,601 x 136)

Line 251-255: the training and testing data are unclear from the authors' description. Neither were they found in the previous section. The second way seems to imply that the authors used "leave-one-out" to do the validation in that training the model by all the data points except one that is tested. However, if this is true, what are the testing data for the first way? Did the authors use all training data for testing, which is definitely incorrect way?

Line 285-289: 6,788 or 6,778 features? Two different numbers appeared. The authors must do a better job to proof-reading before submitting the paper.

Line 295: "...achieved 100.0% cross-validation and 100.0% test accuracy..." I doubt that this statement is true. What is your training/validation data split? After 10-fold cross-validation, how do you use the data to train the final model?

Line 296-297: How were you able to have test accuracy 97.75% and 99.44%, respectively? Line 171 mentioned that there are 1,605 recorded tones and 10% were used for final testing. Thus, either 160 or 161 were the number of testing tones. Either number would not result in 97.75% or 99.44% unless repeated experiments were conducted and average results were reported. In addition, confidence intervals were reported. There must be repeated experiments and is was not mentioned at all in the method.

Line 305: What are "EP front" or "EP Wall"? A figure might be needed for readers to understand.

All figures need to have better quality. Legions are difficult to read, almost unintelligible.

Reviewer #2: This work represents a non-deterministic approach to source separation/localization using machine learning methods. The work likely has merit as proof of concept, but the lack of details make it non-repeatable and I have some concerns about the methodology. Subsequently, I cannot recommend publication as presented. My review largely echoes those of previous reviewers and while I note that there has been some positive movement towards explaining the methods and results in full, much remains to be done before this can be accepted into the bioacoustics and source separation cannon.

A major concern is how the sounds were generated. It appears that the sounds were produced sequentially at each of the speaker locations and the authors used machine learning (ML) to discriminate between these locations. Thus the authors ask, which of these locations could this sound have come from. This is a valid question from a theoretical standpoint but it’s value is only useful when animals are greater than half a body length apart and whistles are not produced in isolation. To make it applicable to the field the authors should ask, ‘Which animal did each source come from’. In doing so the authors would play sources from multiple locations simotaneously. This is especially important given that in ML, it’s difficult to determine which features are the most useful in the discrimination task. If the features are relative amplitude of the first arrival at the four hydrophones then the value of the proposed method is limited.

Technical Concerns

No definition about how tonal extraction was done. Were the 2 second clips taken from 1 after the onset of the tonal recording? From the peak time? From which hydrophone was the timing obtained? For example, they sound clips could be slightly offset on each hydrophone. Did an analysist manually select the clips? How the clips are extracted will determine the features useful for source separation and subsequently what their system uses to discriminate.

Organization

The authors have not organized the Methods section of the manuscript in a manner consistent with the field of bioacoustics and machine learning which is causing considerable confusion. To be consistent with the field, better document their work, and convey their results to the broadest audience possible, the authors should arrange the methods section as follows.

Array setup

This including speaker and receiver positions. Failing to include the hydrophone locations within the body of the manuscript is inexcusable. Where hydrophones are placed relative to the sources and eachother is fundamental to the study and the ability of the methods to generalize. As noted by the previous reviewers, the authors need to include this in the methods as well as in the figure.

Signal generation

This section does not need to be as extensive and could possibly go in supplemental information. One or two paragraphs with the included table will suffice.

The authors choice to exclude harmonics may problematic without considerable explanation. In doing so they have produced idealized tonals which are not representative of biological signals. This choice needs to be justified.

Signal acquisition

This should include how the authors parsed the signals of interest from the recordings. It is not clear how long each recording used to train the regression trees was. Did they collect two seconds around the peak of the received signal? Was it the onset or was some other method used? This is a critical and not well documented aspect of their work.

Further down the authors refer to ‘snipits’ of data. The word choice is fine but what each snipit is and how it was generated needs to go in this section.

Feature Extraction.

This section is key in understanding and replicating their work. As noted by previous reviewers, it’s still not clear. I advise that they exclude or move to the supplemental information any bits of the feature extraction analysis that wasn’t used in the final model and clarify what was included.

Introduction

The value of this work concerns sound source discrimination. The introduction needs to be structured to highlight this. The authors have gone through some efforts to provide background information on the biological motivations for the system but have not done due diligence to the extensive body of work available on source separation in marine mammals. However, there are some issues in what they are presenting in overstating the value of captive studies in signature whistles. While captive studies were integral in the initial study of whistles and have some value in ontogeny initially, the limitations are considerable.

I suggest that the authors re-structure the introduction to focus only on caller discrimination and limit the potential applications to small portions of the discussion. Clarify the issue with source separation in a reverberant environment. Please see work by EM Nosal on sperm whale source separation, a variety of papers by D. Mellenger and other contemporaneous researchers. At minimum, these papers would demonstrate the proper terms in the bioacoustics field.

Methods

86 - The paragraph starting on line 86 is a single sentence. This does not help the reader understand the work.

92 – This paragraph could be considerably simplified

Consider rewording:

We generated 128 unique tonal sounds with pitch and duration within published ranges of Tursiops (table). We used frequency modulated pure tones randomly generated from XXX distribution. For this analysis, harmonics were not included. For details on the generation system see supplemental info.

92- replace ‘sounds’ with ‘tonals’ throughout to refer to your generated signals. ‘sounds’ is too vague for a bioacoustic audience

119- this should be it’s own section (array setup, see above)

120- Consider rewording for clarity:

'The 128 generated signals were played at each of the 14 hydrophone locations corresponding to 7 horizontal positions and two depths, xx m and yy m (Figure). Horizontal hydrophone spacing was approximately XXX between adjacent locations. See linked data below.'

Talking about the cross doesn't add much. Also nix the imperial measurements. Let’s momentarily pretend we are from a civilized country.

123-130- Unclear

Name and thank the assistants in the acknowledgments.

144- ‘Various calibrations’? Define what was calibrated. Explicit details for well established procedures are not needed but I vehemently disagree that defining what they did is ‘outside of the scope of the study’

- First reference to relevant work not included in the manuscript.

145 – Replace ‘collected at’ with ‘sampled at’ as in sample frequency

148- Replaced ‘involved’ with ‘used’

148 – What does ‘Standard Passive acoustic monitoring system’ mean? Does it mean custom matlab scripts were used to autonomously record sound from the hydrophones? As far as I’m aware there are no ‘standard’ methods for PAM in matlab. Clarify.

135- Why is there so much information about the visual recording system? The need for this section is lost on me. Please clarify how the system fits into your study. Knowledge of the hydrophone and speaker locations within a tank shouldn’t require an advanced visual system unless there is considerable and undocumented flow preventing the speakers from remaining stationary. Clarify.

153- This section is the heart of what the authors did and is really difficult to get through. The jargon is very heavy and multiple concepts are introduced in the same sentences. The authors need to walk your readers through this more carefully. There is clearly a lot going on and a lot of work to be carefully discussed. Do justice to it by conveying it to a broader audience.

158- ‘digested’ is the wrong word here. Pick something more accurate.

15-- It's not a, 'so-called' feature set. It’s a feature set. Also, this is part of the confusion. It's not immediately clear what is going on here.

Consider rewording: XXXX features were created from each 4-channel recording of the simulated tonals by converting the wave forms to YYY….

161- GCC-PHAT isn’t explained but TDOA is. The authors need to spend more time on the former and less on the latter.

161 – Currently unpublished -> put it in the supplemental information

Second reference to work not included in the manuscript.

163- Remove ‘briefly’. A ‘brief’ explanation of TDOA consists of , ‘TDOA is the delay in the arrival time of a signal between multiple hydrophones where hydrophones further from the source receive the signal later than those closer to the source’. Feel free to use this.

168- Where ‘elsewhere’? Either this concept is important and warrants explaining and citing or it should be removed.

Third reference to work not included in the manuscr

137-173 a single sentence with multiple interjections. Lost.

Still unsure how you are getting the 120 features from the 16 hydrophones. Hydrophone schematic would help.

176- ‘snipits’- this is reference above and should be defined in your data acquisition.

176-178- this bit is actually quite good and understandable.

179- Would be nice to know what the geometry is…

180 – this section isn’t clear again. Do you mean that the cross correlation time did or did not include second through n-th arrivals?

187- So the only information going into the regression tree is TDOA, cross correlation and GCC-PHAT? Be explicit here to give your readers a break. Remove what didn’t work or add it in the supplemental information.

187 – What do you mean ‘processing’ for each whistle? I thought the snipits were processed for the feature set, not the other way around.

187- This is the first reference to the labeled data. In the ‘feature set’ or earlier in the methods state that you are using supervised learning and your targets and label sets represent the potential source location and XYZ coordinates.

189- Typical phrases in ML to refer to data used to train and test the data are ‘Traning data’, ‘Test Data’, and/or ‘Validation data’. The wording here is awkward

191- what was ‘novel’ about the whistle? Again stick to the same term for the sounds you generated. I suggest, as above, ‘tonal’ and only use ‘whistle’ to refer to the actuall, real life, whistles coming from the animals.

191- This isn’t true, the authors could batch generate but never the less the sentance should go. Consider rewording ‘We chose the Bierman random forest for the classification task due to its ability to reduce the feature space and address whilst performing multi-class classification.

193-196- This is a single sentence with two interjections. Difficult to read consider rewording. The Bierman random forest is a multi-class classifier with a built-in resistance to overfitting through XXXXX. Additionally, the classifier performs feature reduction through YYY.

200- CART acronym without explanation. Write it out the first time.

204 – This remark can easily be read as condescending, especially when taken in context with the various other references to work not included in the paper.

211 – unclear what ‘additional’ models are

221- The localization portion of your study needs it’s own section.

221- replace ‘training sounds’ with ‘snipits’ or ‘tonals’ depending on whether you are referring to the sounds you generated or the sounds you extracted. Consistency.

121-123- Are you referring to the features extracted from each recording? I’m very lost.

222- Three SVM models? In the start of the to-be-created localization section, please provide a brief 1 or 2 sentence overview to SVM and how you use it here.

223- grid point? Be specific throughout.

227- This justification should be in the start of the section about the localization

229- Employed- I hope you paid it a living wage. Replace with ‘used’.

230 – Is the ‘standard’ approach what is referenced in your citation? Or just some details? Clarify.

231- Is this what you mean by grid-point? Note how much later the definition is than it’s first use. If not clarify.

231-236- This could be simplified. Consider:

We divided the available pool space into 6cm grid squares representing all potential source locations. For each grid corner, we calculated the expected TDOA of a source at that location to each of the 16 hydrophones.

233-236- imprecise wording. I know what the authors mean but others may not and all struggle.

237- I assume it’s a single value for soundspeed. State the calculated soundspeed or provide a figure of the soundspeed profile of the pool.

242- state the purpose of the procedure at at the start of the paragraph, not the end.

Results

248- Remove, ‘described above’. Else replace it with a section reference.

252- Not sure what is meant by this.

254- Completely lost. By array you mean the 4 connected hydrophones? The descriptions of when the authors use all 16 hydrophones and when arrays are treated separately isn’t clear to the reader.

248-255- 100% accuracy while increasing the model size strongly suggests overfitting

257-258- This should be in the methods

264- Comma after ‘again’

265- first mention of kriging. Methods.

235- replace ‘test sound’ coordinates with ‘speaker’ or ‘sound source’ coordinates

270- The error looks considerably worse in the Z direction in comparison to the X or Y. This is useful information that I hope is brought up in the discussion

271-277 single sentence

217-281 – This would be useful as a table with rows being the axes and columns as the models

Figure 4- Bar graph or histogram. Remove ‘x-ticks’ denote histogram edges. This is confusing, matlab speak that isn’t useful for non-matlab users.

The figure itself is hard to parse. The variables are the models, the different axes, and the histogram bins. The comparison the authors should be making is the different models (pannals) for each axis. So, I suggest the authors make each axes a different panel and the colors should represent the models. This will make the model comparison significantly easier to see.

Figure 5- I don’t see a lot of value added by this figure.

Figure 5- caption – Remove ‘discussed in the text’. Much of the rest of the caption should be in the methods.

289- removed ‘so-called’,

289-290 – methods.

294- replace ‘ask’ with ‘determine’

298-299- Major point, don’t bury it at the end of the results

Figure 6- Unit labels for colorbars. Replace hydrohpne ‘label’ with hydrophone ‘number’ provide figure of hydrophone layout with each hydrophone number.

Figure 6 label – ‘common panals’ what? First reference. Describe better in ‘array setup’ section to be added

TDOA features do not seem very useful here. Highlight in results.

Discussion

301-303-Run on

307- I presume by ‘sound sources’ you are referring to free-swimming animals? If so say it.

307-310- run on

309-311- unintelligible. Reword.

312 – Are you sure recordings is the word you want? 4-channel arrays/panels? It should be arrays but this needs to be made clear throughout.

314- EP already defined (or should have been)

317- sound ‘source’ originated

321-326- run on

331-333- run on. Start with , ‘Also, it is reassuring that a….’. Try to always put the subject next to the verb or risk sounding like Yoda, do you.

334- Which question? There was no stated question

340- It’s not so much the length of the dolphin that’s important it’s the width. The sound source in a dolphin is near the front of the animal. So, the authors need to highlight that this method shows promise when animals heads are greater than a meter apart which may, or may not, be tenable.

355- Just add the citation after elsewhere remove the rest.

356-Move ‘referring to figure 5’ to later in the sentence. Just reference it as (fig 5)

I think you should replace (EP-Wall) and EP dimensions with X, Y, and Z or lat, lon, depth. Something more intuitive will allow for readers to better understand the results

370- time-of-flight? Used in abstract and elsewhere. Undefined.

371- Reword to say amplitude was not include rather than it was removed. Highlight just TDOA and other methods that were included. Consider, ‘In this work TDOA and Cross correlation values were used to discriminate between source locations. Direct or relative amplitude was not included in the feature set’. Or something like that.

Acknowledgments

This is an online journal without word limits. Name and graciously thank the two-dozen people who helped you. Don’t be lazy.

6. PLOS authors have the option to publish the peer review history of their article (what does this mean?). If published, this will include your full peer review and any attached files.

Reviewer #1: No

Reviewer #2: No

---

## [Author Response · Author response to Decision Letter 0]

21 Feb 2020

Dear Dr. Matsumoto and Reviewers,

We thank you for the time and patience you have dedicated to reading and thoroughly, constructively criticizing our paper. In almost all cases we have striven to follow your direction, and we hope that the results please you. To a few, mostly minor, points we do raise objections and explain ourselves. Most notably, we do not agree with Reviewer 2’s suggestion to strike our motivating application of the proposed methods, namely whistle localization for dolphins in an enclosed environment, from the Introduction and generally the paper. We feel this application is inextricably linked to our study design. In any case, we respond to this and all other specific comments below.

Thank you,

Sean Woodward

Diana Reiss

Marcelo Magnasco

Responses to Editor and Reviewer comments are given in bold red (only in PDF version).

PONE-D-19-33956

Machine Source Localization of Tursiops truncatus Whistle-like Sounds in a Reverberant Aquatic Environment

PLOS ONE

Dear Dr. Woodward,

Thank you for submitting your manuscript to PLOS ONE. After careful consideration, we feel that it has merit but does not fully meet PLOS ONE’s publication criteria as it currently stands. Therefore, we invite you to submit a revised version of the manuscript that addresses the points raised during the review process.

We would appreciate receiving your revised manuscript by Feb 23 2020 11:59PM. To enhance the reproducibility of your results, we recommend that if applicable you deposit your laboratory protocols in protocols.io, where a protocol can be assigned its own identifier (DOI) such that it can be cited independently in the future. For instructions see: http://journals.plos.org/plosone/s/submission-guidelines#loc-laboratory-protocols

A rebuttal letter that responds to each point raised by the academic editor and reviewer(s). This letter should be uploaded as separate file and labeled 'Response to Reviewers'.

A marked-up copy of your manuscript that highlights changes made to the original version. This file should be uploaded as separate file and labeled 'Revised Manuscript with Track Changes'.

An unmarked version of your revised paper without tracked changes. This file should be uploaded as separate file and labeled 'Manuscript'.

We look forward to receiving your revised manuscript.

Kind regards,

Haru Matsumoto

Academic Editor

PLOS ONE

Journal Requirements:

'We thank the National Aquarium for participating in this study, as well the National 402

Science Foundation (Awards 1530544, 1607280), the Eric and Wendy Schmidt Fund for 403

Strategic Innovation, and the Rockefeller University for funding.'

'MO Magnasco, DR Reiss

Awards 530544, 1607280

National Science Foundation

https://www.nsf.gov

The funders had no role in study design, data collection and analysis, decision to

publish, or preparation of the manuscript.'

Additional Editor Comments (if provided):

Dear Dr. Woodward,

The manuscript has improved but it is still lacking details and scientific discussions. Although the application of machine learning to animal localization is unique and the results are interesting, I have to agree with both reviewers that the manuscript needs a major revision. As guidelines, 1) as reviewer 2 pointed out, PLOS One draws a broader audience and not just for experts in machine learning or bio-acoustics. For that, it needs more explanation without losing your audience. 2) The experiment must be repeatable with details of experimental set-up (e.g., hydrophone locations in XYZ as pointed out by both reviewers). I am sure that marine bio-acoustics researchers are eager to apply your ML method to the other animals in the ocean if you can describe the experiment clearly with a few jargon. Unlike the ocean, the aquarium tank is a controlled environment. 3) Re-submission without the details of your aquarium experiment set up and methodology, especially the dimensional information, will jeopardize further review. 4) Before resubmitting the revision, please ask your coauthors to proofread your manuscript carefully in order to save the reviewer's time.

Regards,

Haru Matsumoto

(1) Per Reviewer 2’s detailed comments, we have added more explanation of our methods. While we have also provided more background on pre-established methods, unless specifically requested by Reviewer 2 we were sometimes unclear as to where more background is desired, and how much. The paper makes mention of many pre-established methods (that are arguably not expert-level machine learning or bioacoustics, such as the decision tree), and we question whether thoroughly introducing the reader to them all is what’s desired and/or appropriate. That being said, again we have added considerable background, and we continue to be open to adding more.

(2, 3) We have added this information.

(4) This has been done.

Reviewers' comments:

Reviewer's Responses to Questions

Comments to the Author

1. Is the manuscript technically sound, and do the data support the conclusions?

Reviewer #1: Partly

Reviewer #2: Partly

2. Has the statistical analysis been performed appropriately and rigorously? 

Reviewer #1: No

Reviewer #2: Yes

3. Have the authors made all data underlying the findings in their manuscript fully available?

Reviewer #1: Yes

Reviewer #2: Yes

4. Is the manuscript presented in an intelligible fashion and written in standard English?

Reviewer #1: No

Reviewer #2: No

5. Review Comments to the Author

Reviewer #1: This paper aims to tackle an important challenge of localization: associate the recorded whistle to the sound-producing animal. Instead of TDOA-based localization algorithms, the authors used classify the approximate locations and estimate the locations (regression) through random forest, support vector machine and other machine learning algorithms. It was a promising work but the data could be summarized more clearly, experiment setup laid out more precisely and results reported more systematically. It seems to be a work written by a junior academic member without proof-reading from other co-authors. There're simply too many errors and inconsistency all over the manuscript. I recognized that there might be a diamond in the rough. However, the paper have to be organized in a better shape so that scientific findings can be communicated effectively.

Detailed comments are as follows:

In abstract, please refrain from using abbreviations without the full names. What is MAD? What is IQR? Why is SRP the abbreviation of Steered-Response? What does P stand for?

It's not clear what it means by "...comparable accuracy - even when interpolating at several meters - in the lateral directions when deprived of training data at testing sites..." and "...in all domains..."

All abbreviations in the Abstract are now defined. We have reworded the troublesome lines. 

Line 120-123: It'll be easier to understand through an equation or an example on the figure.

An equation has been added.

Line 127: It's almost impossible to imagine what y(t) might look like. The denominator arcsin(Phi(t)) are very likely to be incorrect since the input to arcsin needs to be [-1, 1] whereas Phi(t) can be outside this range.

A mistake in the denominator has been corrected; it resulted from a source material conflict. We apologize for the error.

Line 129: Why does m=0.8 correspond to two harmonics?

We have added more detail about the equation, being more explicit about the mechanism through which harmonics are added (specifically, the triangularization of the underlying waveform) and the fact that harmonics successively decrease in signal power. As we write, at m=0.8 we do not see a third harmonic above noise in practice.

Line 133: What is "pre-speaker"?

Word choice has been changed: “…example of a tonal generated by the above procedure…”

In the legend of Figure 1, it says duration = 1 s whereas Figure 1 only shows roughly 340 ms. In addition, please use SI units everywhere in the manuscript, i.e., "s" instead of "second" or "sec". Details and examples of SI units can be seen:

https://www.bipm.org/en/measurement-units/

The bad duration (from a previous, different choice of tonal) has been fixed. 

We now use SI units everywhere. 

Line 151: what are the depths of the hydrophone? What's the separation of the four hydrophones in a location? If there's no significant separation of depth, how can you expect that they are able to locate the depth of locations?

Hydrophone coordinates are now given, with their relationship to the pool surface provided in the text. Hydrophones are approximately 1.60 +/- 0.22 m under the surface. Yes, we expected decreased resolution in the direction of depth (theoretically these separations are not necessarily limiting), but how much depended on the particulars of the sounds and acoustic environment.

Line 163: the authors mentioned "Audacity AUP sound format". AUP is not sound format. It's Audacity's format for organizing projects.

This has been clarified.

Line 175: "...with sinusoids and quasi-sinusoids with the same parameters grouped together given their close similarity." There might be too many "with"s.

Fixed.

Line 205 - 207: Why does 27,126 Fourier transform elements correspond to 0 to 27,126 Hz? What is your FFT/DFT size? In order to have 1 s frequency resolution, DFT size needs to be 192,000!

Per Reviewer 2’s comments, we have decided to eliminate discussion of these features, as they ultimately went entirely unused. But, yes, we used large windows to eliminate time dependence.

Line 210: Why is the number of features 897,871? I calculated it to be 897,736 (=6,601 x 136)

Your number is correct, ours was not updated. We apologize.

Line 251-255: the training and testing data are unclear from the authors' description. Neither were they found in the previous section. The second way seems to imply that the authors used "leave-one-out" to do the validation in that training the model by all the data points except one that is tested. However, if this is true, what are the testing data for the first way? Did the authors use all training data for testing, which is definitely incorrect way?

We’ve clarified our methods. Except for the random forest (for which OOB error was used), all validation error was determined by 10-fold cross validation on the training/validation data. It would have been reasonable at this stage to perform hyperparameter optimization, but in essentially every case we didn’t, either because the model class doesn’t heavily rely on it and/or because we didn’t want to further promote overfitting (in the case of the basic classification tree) and were content with off-the-shelf prediction. We are unclear about the meaning of the reviewer’s last sentence. We didn’t include testing data among our training/validation data. However, we did aggregate all training/validation data into final models before evaluation on the test data, but this is correct.

Line 285-289: 6,788 or 6,778 features? Two different numbers appeared. The authors must do a better job to proof-reading before submitting the paper.

The latter. We apologize, it has been corrected.

Line 295: "...achieved 100.0% cross-validation and 100.0% test accuracy..." I doubt that this statement is true. What is your training/validation data split? After 10-fold cross-validation, how do you use the data to train the final model?

We used a 9:1 training/validation:test set split. For test-set evaluation, all data from the training/validation set were used to train the model. Hyperparameter optimization was not performed with evaluation on the test set, and actually not performed on the training/validation set either (though regression model selection was, which we now state explicitly); LDA has no hyperparameters, we pre-specified the main hyperparameter choice for the SVM’s (kernel class), and default hyperparameters were kept for the classification tree for fear of over-fitting.

Line 296-297: How were you able to have test accuracy 97.75% and 99.44%, respectively? Line 171 mentioned that there are 1,605 recorded tones and 10% were used for final testing. Thus, either 160 or 161 were the number of testing tones. Either number would not result in 97.75% or 99.44% unless repeated experiments were conducted and average results were reported. In addition, confidence intervals were reported. There must be repeated experiments and is was not mentioned at all in the method.

We profusely apologize, the training/validation set size of 1,605 was given where the total of 1,783 should have been. The 10% testing size was thus 178 and 97.75% and 99.44% correspond to 174 and 178 tonals, respectively.

Line 305: What are "EP front" or "EP Wall"? A figure might be needed for readers to understand.

The now-updated axis names are included in two figures now.

All figures need to have better quality. Legions are difficult to read, almost unintelligible.

The quality of our figures was/is significantly degraded during the PLOS reviewer manuscript creation process. We confirmed this with PLOS help staff, which states that at this stage the figures must merely be high-quality enough for content review. We apologize for any inconvenience this causes, and almost everywhere have provided higher-resolution figures to try to compensate for review.

Reviewer #2: This work represents a non-deterministic approach to source separation/localization using machine learning methods. The work likely has merit as proof of concept, but the lack of details make it non-repeatable and I have some concerns about the methodology. Subsequently, I cannot recommend publication as presented. My review largely echoes those of previous reviewers and while I note that there has been some positive movement towards explaining the methods and results in full, much remains to be done before this can be accepted into the bioacoustics and source separation cannon.

A major concern is how the sounds were generated. It appears that the sounds were produced sequentially at each of the speaker locations and the authors used machine learning (ML) to discriminate between these locations. Thus the authors ask, which of these locations could this sound have come from. This is a valid question from a theoretical standpoint but it’s value is only useful when animals are greater than half a body length apart and whistles are not produced in isolation. To make it applicable to the field the authors should ask, ‘Which animal did each source come from’. In doing so the authors would play sources from multiple locations simotaneously. This is especially important given that in ML, it’s difficult to determine which features are the most useful in the discrimination task. If the features are relative amplitude of the first arrival at the four hydrophones then the value of the proposed method is limited.

The reviewer suggests interesting avenues for investigation, however we do feel that it is valid to consider the localization of whistles that do not occur simultaneously. This is a realistic problem, at least in the captive setting where vocalizations are often sparse. Also, we note it remains unclear whether our limited training set demonstrates the best potential accuracy available to the machine learning models considered here. 

Moreover, because we suggest methods of source localization based on whistle fundamentals alone, there exists the possibility of applying these methods to simultaneously occurring whistles whose fundamentals can be satisfactorily separated by filtering. Of course, we admit there still exists the opportunity to extend these methods to overlapping traces.

The problem of classifying sounds by speaker identity rather than by source location is fascinating (having only been explored only for signature whistles to our knowledge), however addressing it is not currently within our reach. We have imagined using the present source localization methods to aid in the whistle attribution for dolphins at the National Aquarium in order to create a large training set suited to this purpose. But fate has been cruel to our experimental setup and we must now leave this task to others.

Technical Concerns

No definition about how tonal extraction was done. Were the 2 second clips taken from 1 after the onset of the tonal recording? From the peak time? From which hydrophone was the timing obtained? For example, they sound clips could be slightly offset on each hydrophone. Did an analysist manually select the clips? How the clips are extracted will determine the features useful for source separation and subsequently what their system uses to discriminate.

We have added these details (Line ~281), within the new Methods structure the reviewer suggested. We also discuss how our choice of features was envisioned to treat our automatically-extracted tonals the same as manually-extracted whistles (i.e., by not assuming fixed times between window starts and tonal onsets).

Organization

The authors have not organized the Methods section of the manuscript in a manner consistent with the field of bioacoustics and machine learning which is causing considerable confusion. To be consistent with the field, better document their work, and convey their results to the broadest audience possible, the authors should arrange the methods section as follows.

Array setup

This including speaker and receiver positions. Failing to include the hydrophone locations within the body of the manuscript is inexcusable. Where hydrophones are placed relative to the sources and eachother is fundamental to the study and the ability of the methods to generalize. As noted by the previous reviewers, the authors need to include this in the methods as well as in the figure.

Signal generation

This section does not need to be as extensive and could possibly go in supplemental information. One or two paragraphs with the included table will suffice.

The authors choice to exclude harmonics may problematic without considerable explanation. In doing so they have produced idealized tonals which are not representative of biological signals. This choice needs to be justified.

Signal acquisition

This should include how the authors parsed the signals of interest from the recordings. It is not clear how long each recording used to train the regression trees was. Did they collect two seconds around the peak of the received signal? Was it the onset or was some other method used? This is a critical and not well documented aspect of their work.

Further down the authors refer to ‘snipits’ of data. The word choice is fine but what each snipit is and how it was generated needs to go in this section.

Feature Extraction.

This section is key in understanding and replicating their work. As noted by previous reviewers, it’s still not clear. I advise that they exclude or move to the supplemental information any bits of the feature extraction analysis that wasn’t used in the final model and clarify what was included.

We have executed the recommended restructuring.

Hydrophone coordinates are now provided (Table 1).

The exclusion of whistle harmonics is now discussed around Line 220. The main justification is that we did not want to make unnecessary assumptions about the fundamental-harmonic relationships during signal generation (this is why we ultimately decided to remove the faux-harmonics from the triangular waveform tonals), when with appropriate filtering the problem of localizing real whistles should be reducible to the problem of localizing fundamentals, given their frequency separation. As we see it, the harmonics simply offer extra information that could potentially help otherwise successful models.

Introduction

The value of this work concerns sound source discrimination. The introduction needs to be structured to highlight this. The authors have gone through some efforts to provide background information on the biological motivations for the system but have not done due diligence to the extensive body of work available on source separation in marine mammals. However, there are some issues in what they are presenting in overstating the value of captive studies in signature whistles. While captive studies were integral in the initial study of whistles and have some value in ontogeny initially, the limitations are considerable.

I suggest that the authors re-structure the introduction to focus only on caller discrimination and limit the potential applications to small portions of the discussion. Clarify the issue with source separation in a reverberant environment. Please see work by EM Nosal on sperm whale source separation, a variety of papers by D. Mellenger and other contemporaneous researchers. At minimum, these papers would demonstrate the proper terms in the bioacoustics field.

The present research project’s design is highly particular to the study of dolphin whistles in an enclosed environment; from our assumptions about the “sound space” and our restrictive choice of training sounds (sinusoidal narrowband sounds), to our assumption that sound sources are acoustically identical (which dolphins tentatively are with respect to non-signature whistles), to our choice of environment (a concrete dolphin tank), to the choice of sound localization methods (which require spatial sampling, practical mainly for relatively small environments), the design is specific to our proposed application. While we hope that the methods proposed here have other applications, our study was not designed with them in mind, and -- partly given the black box nature of machine learning -- we are not positioned to predict to which of these more distant applications our results might be relevant. Similarly, among all potential applications (especially within the area of “caller discrimination”) we suspect our study is of primary importance to the particular biological application we propose, which we feel is indeed valid. We feel that it would be unfair to the authors to deprive the paper of this relevance. While, alternatively, the reviewer may suggest we frame this project as a purely abstract exercise, we feel we do have sufficient justification for equating our design to this real and valid application, the whistle source localization for bottlenose dolphins in an enclosed environment.

Studying the vocalizations of captive dolphins has limitations, yes, but also presents special opportunities. Rarely in other contexts is it possible monitor a fixed group of individuals continuously for months at a time or longer -- this is still not possible with dolphin dtags. Moreover, rarely in other contexts is training and psychological/behavioral experimentation possible, which include (humane) methods that have proven uniquely helpful for studying dolphin cognition and language comprehension. We broadly refer to the studies of the late Louis Herman and humbly opine that much of his aquaria research could not realistically be replicated in the field at this time. As reflected by the sophistication of the field of birdsong research and its relationship to captive studies, we feel that responsible captive studies serve an established scientific role in the study of language and related cognition, that cannot be discarded simply for the loss of some wild behavior. Of course, captive research’s relationship to animal welfare is an important topic worthy of consideration. However, as we are merely addressing an ancillary problem to captive research (a problem that may also extend to purely passive research on animals in a sanctuary environment), we feel that at present this topic is not relevant.

We await more feedback on the above. In the meantime, we are happy to continue to make our methods understandable and recognizable within the broader field of marine mammal communication. To this end, we have significantly expanded our Introduction to include more background information about sound source separation/attribution, with special attention paid to marine mammalogy. We do try to convey that our goals deviate slightly from those of sound source separation as described in other contexts.

Methods

86 - The paragraph starting on line 86 is a single sentence. This does not help the reader understand the work.

The one sentence was broken into two, with clearer separation between clauses.

92 – This paragraph could be considerably simplified

Consider rewording:

We generated 128 unique tonal sounds with pitch and duration within published ranges of Tursiops (table). We used frequency modulated pure tones randomly generated from XXX distribution. For this analysis, harmonics were not included. For details on the generation system see supplemental info.

We thank the reviewer for the suggested wording, and have incorporated it into our new version.

92- replace ‘sounds’ with ‘tonals’ throughout to refer to your generated signals. ‘sounds’ is too vague for a bioacoustic audience

We now refer to our generated/created sounds as “tonals.”

119- this should be it’s own section (array setup, see above)

Methods have been restructured.

120- Consider rewording for clarity:

'The 128 generated signals were played at each of the 14 hydrophone locations corresponding to 7 horizontal positions and two depths, xx m and yy m (Figure). Horizontal hydrophone spacing was approximately XXX between adjacent locations. See linked data below.'

Talking about the cross doesn't add much. Also nix the imperial measurements. Let’s momentarily pretend we are from a civilized country.

Recommendations have been adopted.

123-130- Unclear

Name and thank the assistants in the acknowledgments.

This is indeed appropriate. The main author concedes his negligence and thanks the reviewer for this especially important correction.

144- ‘Various calibrations’? Define what was calibrated. Explicit details for well established procedures are not needed but I vehemently disagree that defining what they did is ‘outside of the scope of the study’

- First reference to relevant work not included in the manuscript.

Clarification added. We make it explicit that the localization methods proposed in this study did not make use of measured hydrophone coordinates (which motivated our previous comment about scope). In light of this, we opted to remove mention of the work we previously indicated as relevant instead of citing it, because it has not yet undergone peer review; we would be happy to reintroduce the mention with a citation if the editor and reviewer prefer.

145 – Replace ‘collected at’ with ‘sampled at’ as in sample frequency

Done.

148- Replaced ‘involved’ with ‘used’

Done.

148 – What does ‘Standard Passive acoustic monitoring system’ mean? Does it mean custom matlab scripts were used to autonomously record sound from the hydrophones? As far as I’m aware there are no ‘standard’ methods for PAM in matlab. Clarify.

We were referring to passive operation of the sound recording system (i.e., for multi-day/week recordings), which we distinguish from the operator-centric, Audacity-based mode of operation used during this study. We have opted to delete mention of passive system operation for clarity.

135- Why is there so much information about the visual recording system? The need for this section is lost on me. Please clarify how the system fits into your study. Knowledge of the hydrophone and speaker locations within a tank shouldn’t require an advanced visual system unless there is considerable and undocumented flow preventing the speakers from remaining stationary. Clarify.

The extent of the camera system is not relevant to this study. We have removed mention of the total camera count along with mention of the camera make/model from the text body. We have left make/model information in the image caption, where we feel it is not out of place. 

153- This section is the heart of what the authors did and is really difficult to get through. The jargon is very heavy and multiple concepts are introduced in the same sentences. The authors need to walk your readers through this more carefully. There is clearly a lot going on and a lot of work to be carefully discussed. Do justice to it by conveying it to a broader audience.

We have expanded this section, including more explanation. We acknowledge that PLOS One appeals to a broad audience, at the same time we note that this paper has a lot of ground to cover and are unsure how much explanation is warranted. We can open up this section further should the reviewers like.

158- ‘digested’ is the wrong word here. Pick something more accurate.’

This word and sentence has been removed as part of a larger rewrite.

15-- It's not a, 'so-called' feature set. It’s a feature set. Also, this is part of the confusion. It's not immediately clear what is going on here.

Consider rewording: XXXX features were created from each 4-channel recording of the simulated tonals by converting the wave forms to YYY….

The recommendation has been incorporated into a rewrite. 

161- GCC-PHAT isn’t explained but TDOA is. The authors need to spend more time on the former and less on the latter.

We have expanded the explanation of GCC-PHAT.

161 – Currently unpublished -> put it in the supplemental information

Second reference to work not included in the manuscript.

This reference has been removed.

163- Remove ‘briefly’. A ‘brief’ explanation of TDOA consists of , ‘TDOA is the delay in the arrival time of a signal between multiple hydrophones where hydrophones further from the source receive the signal later than those closer to the source’. Feel free to use this.

A much briefer definition of the TDOA has been included elsewhere as part of a larger rewrite.

168- Where ‘elsewhere’? Either this concept is important and warrants explaining and citing or it should be removed.

Third reference to work not included in the manuscr

This reference has been removed.

137-173 a single sentence with multiple interjections. Lost.

Still unsure how you are getting the 120 features from the 16 hydrophones. Hydrophone schematic would help.

The troublesome sentence has been removed as part of a rewrite.

More information about hydrophone geometry has been provided, in the new Hydrophone Setup subsection.

We now share the calculation of the number of unique hydrophone pairs in 16, namely “16 choose 2” = 16! /(2! x 14!) = 120.

176- ‘snipits’- this is reference above and should be defined in your data acquisition.

We have defined snippets.

176-178- this bit is actually quite good and understandable.

Thank you!

179- Would be nice to know what the geometry is…

Now included in the new Hydrophone Setup subsection.

180 – this section isn’t clear again. Do you mean that the cross correlation time did or did not include second through n-th arrivals?

We expanded this section. What we’re saying is that we truncated the cross-correlations generously enough to ensure inclusion of first-arrival TDOA information for any given tonal between any given pair of hydrophones. TDOA information for some pairs of later arrivals is included, we just do not make specific guarantees. And note that pairs of later arrivals do not necessarily correspond to larger TDOA’s than the first-arrival TDOA’s. In fact, pairs of later arrivals can correspond to smaller TDOA’s. Take, for instance, a complex multi-path to a nearby hydrophone versus a simple multi-path to a distant hydrophone, for some source location; these higher-order paths can be imagined to be arbitrarily similar in length and to correspond to a TDOA arbitrarily close to zero. 

187- So the only information going into the regression tree is TDOA, cross correlation and GCC-PHAT? Be explicit here to give your readers a break. Remove what didn’t work or add it in the supplemental information.

Done.

187 – What do you mean ‘processing’ for each whistle? I thought the snipits were processed for the feature set, not the other way around.

The use of “processing” in this way has been removed as part of a larger rewrite.

187- This is the first reference to the labeled data. In the ‘feature set’ or earlier in the methods state that you are using supervised learning and your targets and label sets represent the potential source location and XYZ coordinates.

The word is now introduced earlier (Line 316).

189- Typical phrases in ML to refer to data used to train and test the data are ‘Traning data’, ‘Test Data’, and/or ‘Validation data’. The wording here is awkward

Introduced relevant terminology earlier, in part to avoid awkward wording here.

191- what was ‘novel’ about the whistle? Again stick to the same term for the sounds you generated. I suggest, as above, ‘tonal’ and only use ‘whistle’ to refer to the actuall, real life, whistles coming from the animals.

Use of “novel” as in “novel whistle” has been deemed unnecessary and removed in most cases. We were referring to tonals that were not used in a model’s training, but were not necessarily in the test set. We continue to use “novel” with respect to tonals from un-trained source locations, but are clearer about this meaning.

The suggested change in terminology has been implemented.

191- This isn’t true, the authors could batch generate but never the less the sentance should go. Consider rewording ‘We chose the Bierman random forest for the classification task due to its ability to reduce the feature space and address whilst performing multi-class classification.

While the original sentence was overly dramatic (and removed), we ultimately stand by it; while the authors are aware of batch training, an excessively wide feature set is not as aptly addressed by batch training as an excessively deep one, for which the method is ultimately intended.

The writers have reduced the other sentence for clarity.

193-196- This is a single sentence with two interjections. Difficult to read consider rewording. The Bierman random forest is a multi-class classifier with a built-in resistance to overfitting through XXXXX. Additionally, the classifier performs feature reduction through YYY.

The sentence has been reworded for clarity.

200- CART acronym without explanation. Write it out the first time.

Done.

204 – This remark can easily be read as condescending, especially when taken in context with the various other references to work not included in the paper.

We have rephrased this remark.

211 – unclear what ‘additional’ models are

Hopefully this has been clarified.

221- The localization portion of your study needs it’s own section.

Done.

221- replace ‘training sounds’ with ‘snipits’ or ‘tonals’ depending on whether you are referring to the sounds you generated or the sounds you extracted. Consistency.

We restrict ourselves to these two terms now. 

Done.

121-123- Are you referring to the features extracted from each recording? I’m very lost.

This section has largely been rewritten.

222- Three SVM models? In the start of the to-be-created localization section, please provide a brief 1 or 2 sentence overview to SVM and how you use it here.

Given the line referenced and the mention of three models we think the author is referring to our Gaussian process regression models. We have therefore included more information about Gaussian process regression. 

223- grid point? Be specific throughout.

We removed references to grid points (outside of the SRP-PHAT section).

227- This justification should be in the start of the section about the localization

We have considered this comment, however it seems to us that talking about the limitations of interpolative regression on our test data before we’ve explained the interpolative regression may confuse the reader. Also, we put regression after classification because it seems like the natural order of problem difficulty.

229- Employed- I hope you paid it a living wage. Replace with ‘used’.

Done.

230 – Is the ‘standard’ approach what is referenced in your citation? Or just some details? Clarify.

We have expanded/clarified “standard methodology” per this comment.

231- Is this what you mean by grid-point? Note how much later the definition is than it’s first use. If not clarify.

The ambiguity has been resolved by limiting references of “grid points” to this section.

231-236- This could be simplified. Consider:

We divided the available pool space into 6cm grid squares representing all potential source locations. For each grid corner, we calculated the expected TDOA of a source at that location to each of the 16 hydrophones.

Thank you for the recommendation, we pulled from it for our rewrite.

233-236- imprecise wording. I know what the authors mean but others may not and all struggle.

We are now more precise and explicit in our explanation.

237- I assume it’s a single value for soundspeed. State the calculated soundspeed or provide a figure of the soundspeed profile of the pool.

The value is now given.

242- state the purpose of the procedure at at the start of the paragraph, not the end.

Done. Also, the sentence has been rephrased.

Results

248- Remove, ‘described above’. Else replace it with a section reference.

Done.

252- Not sure what is meant by this.

The sentence has been reworded for clarity.

254- Completely lost. By array you mean the 4 connected hydrophones? The descriptions of when the authors use all 16 hydrophones and when arrays are treated separately isn’t clear to the reader.

In all instances we now refer to our hydrophone arrays as “4-hydrophone-arrays.”

248-255- 100% accuracy while increasing the model size strongly suggests overfitting

The authors do not fully understand this criticism. Adding more trees to a random forest to minimize validation error is standard practice. As an ensemble method in which the trees, each trained on a subset of features, “vote” on the classification of each sample, adding trees is not anticipated to increase overfitting. https://arxiv.org/abs/1407.7502 In any case, checking for overfitting is the purpose of evaluation on the held-out test set. Moreover, that our linear models achieved similar results also suggests against overfitting. 

257-258- This should be in the methods

Yes, done.

264- Comma after ‘again’

Done.

265- first mention of kriging. Methods.

This was in methods, however we’ve added more description with regard to our motivation to use the method.

235- replace ‘test sound’ coordinates with ‘speaker’ or ‘sound source’ coordinates

We now refer primarily to “source coordinates of tonals.”

270- The error looks considerably worse in the Z direction in comparison to the X or Y. This is useful information that I hope is brought up in the discussion

It is! Yes, we think it is important as well.

271-277 single sentence

The sentence has been split for clarity.

217-281 – This would be useful as a table with rows being the axes and columns as the models

Figure 4- Bar graph or histogram. Remove ‘x-ticks’ denote histogram edges. This is confusing, matlab speak that isn’t useful for non-matlab users.

The figure itself is hard to parse. The variables are the models, the different axes, and the histogram bins. The comparison the authors should be making is the different models (pannals) for each axis. So, I suggest the authors make each axes a different panel and the colors should represent the models. This will make the model comparison significantly easier to see.

We have rearranged the plots as recommended; we now have one for each axis. We have removed the reference to the ticks.

Figure 5- I don’t see a lot of value added by this figure.

This figure compares the overall accuracy of the different methods. We feel this graph summarizes a main finding of the paper -- the comparison of Gaussian regression with SRP -- in a statistically correct way that the table of MAD’s and histograms do not, and in a way that we not execute as succinctly in text.

Figure 5- caption – Remove ‘discussed in the text’. Much of the rest of the caption should be in the methods.

Done.

289- removed ‘so-called’,

Done.

289-290 – methods.

This was already included in Methods, but is now more prominent.

294- replace ‘ask’ with ‘determine’

Done.

298-299- Major point, don’t bury it at the end of the results

We now highlight this result in the Abstract and Introduction.

Figure 6- Unit labels for colorbars. Replace hydrohpne ‘label’ with hydrophone ‘number’ provide figure of hydrophone layout with each hydrophone number.

Done.

Figure 6 label – ‘common panals’ what? First reference. Describe better in ‘array setup’ section to be added

TDOA features do not seem very useful here. Highlight in results.

We now refer to “4-hydrophone-arrays” consistently. The Hydrophone Setup subsection has been added.

Discussion

301-303-Run on

The sentence has been split for clarity.

307- I presume by ‘sound sources’ you are referring to free-swimming animals? If so say it.

Done.

307-310- run on

The sentence has been split for clarity.

309-311- unintelligible. Reword.

The sentence has been reworded for clarity.

312 – Are you sure recordings is the word you want? 4-channel arrays/panels? It should be arrays but this needs to be made clear throughout.

We now refer to “WAV samples” and “snippets” (previously defined) here.

We now universally refer to arrays/panels as “4-hydrophone-arrays” for consistency and clarity.

314- EP already defined (or should have been)

This was a reminder. Re-definition has been removed, however.

317- sound ‘source’ originated

We now universally refer to “[sound] source locations.”

321-326- run on

The sentence has been split for clarity.

331-333- run on. Start with , ‘Also, it is reassuring that a….’. Try to always put the subject next to the verb or risk sounding like Yoda, do you.

We humbly suggest that Yoda’s ears suggest a level of expertise in the area of sound source localization that make him worth evoking. 

Fixed!

334- Which question? There was no stated question

We have rephrased this.

340- It’s not so much the length of the dolphin that’s important it’s the width. The sound source in a dolphin is near the front of the animal. So, the authors need to highlight that this method shows promise when animals heads are greater than a meter apart which may, or may not, be tenable.

Although we continue to reference dolphin body length as a crude yardstick, we acknowledge the importance of the reviewer’s point and have indicated that our method’s error is greater than the minimum possible separation between two dolphins’ heads.

355- Just add the citation after elsewhere remove the rest.

Done.

356-Move ‘referring to figure 5’ to later in the sentence. Just reference it as (fig 5)

I think you should replace (EP-Wall) and EP dimensions with X, Y, and Z or lat, lon, depth. Something more intuitive will allow for readers to better understand the results

The reference has been moved.

We now use X, Y, Z for pool axis names.

370- time-of-flight? Used in abstract and elsewhere. Undefined.

We no longer reference time-of-flight.

371- Reword to say amplitude was not include rather than it was removed. Highlight just TDOA and other methods that were included. Consider, ‘In this work TDOA and Cross correlation values were used to discriminate between source locations. Direct or relative amplitude was not included in the feature set’. Or something like that.

Done.

Acknowledgments

This is an online journal without word limits. Name and graciously thank the two-dozen people who helped you. Don’t be lazy.

Yes, we recognize our error and gladly include all names!

6. PLOS authors have the option to publish the peer review history of their article (what does this mean?). If published, this will include your full peer review and any attached files.

Do you want your identity to be public for this peer review? For information about this choice, including consent withdrawal, please see our Privacy Policy.

Reviewer #1: No

Reviewer #2: No

---

## [Editor Report · Decision Letter 1]

2 Mar 2020

PONE-D-19-33956R1

Machine Source Localization of *Tursiops truncatus* Whistle-like Sounds in a Reverberant Aquatic Environment

PLOS ONE

Dear Dr. Woodward,

Thank you for submitting your manuscript to PLOS ONE. After careful consideration, we feel that it has merit but does not fully meet PLOS ONE’s publication criteria as it currently stands. Therefore, we invite you to submit a revised version of the manuscript that addresses the points raised during the review process.

We would appreciate receiving your revised manuscript by Apr 16 2020 11:59PM. To enhance the reproducibility of your results, we recommend that if applicable you deposit your laboratory protocols in protocols.io, where a protocol can be assigned its own identifier (DOI) such that it can be cited independently in the future. For instructions see: http://journals.plos.org/plosone/s/submission-guidelines#loc-laboratory-protocols

We look forward to receiving your revised manuscript.

Kind regards,

Haru Matsumoto

Academic Editor

PLOS ONE

Additional Editor Comments (if provided):

Dear Dr. Woodward,

I believe that it is the first application of ML to the localization problem of the sound sources in the water. Your research is unique and results are interesting. You have improved the manuscript by taking suggestions from the two reviewers. All the errors pointed out by the reviewers' were corrected. The experimental setup and results are more clear now. However, the main criticism of the two reviewers is the intelligibility of the manuscript, which is still a problem. For example, the abstract and intro are very confusing and difficult to follow even for us in underwater acoustics and marine mammal fields. Please keep in mind that often the abstract and the results are the sections that readers read first. If they are hard to follow, readers would not read the rest and would not cite your research. The reviewers did the job they were asked and I am not going to send the manuscript back to them for further review. I have suggested in my previous comment to invite someone outside of ML or AI field (preferably someone with underwater acoustic background) to read the manuscript carefully and improve intelligibility. Have you done that? You can include him/her as a last co-author.

If you disagree with my decision, another option for you is to request PLOS ONE to change the science editor, which I do not mind at all. But please keep in mind that it would be the same long process again of finding new reviewers and revising.

One error that I just noticed is in Fig. 2. The unit you use, dB/Hz is not correct. It should be dB relative to the standard unit (Pa/sqrt(Hz) for sound). Also it is not clear if it is the actual sound, electrical signal or simulation. You have used SQ26-08 hydrophone from Cetacean Research, which is a calibrated hydrophone. But the dB values are too small for sound or electrical power. Regardless, the spectral intensity is already normalized by a unit frequency (uPa/sqrt(Hz) for sound) when you do FFT. Also, it may be interesting for readers if you can include one of the hydrophone channel signals and spectrograms in the reverberated environment of the tank.

Regards,

Haru Matsumoto
---

## [Author Response · Author response to Decision Letter 1]

3 Jun 2020

(No additional referee reviews were included in this revision)

---

## [Editor Report · Decision Letter 2]

10 Jun 2020

Learning to localize sounds in a highly reverberant environment: machine-learning tracking of dolphin whistle-like sounds in a pool

PONE-D-19-33956R2

Dear Dr. Magnasco,

We’re pleased to inform you that your manuscript has been judged scientifically suitable for publication and will be formally accepted for publication once it meets all outstanding technical requirements.

Kind regards,

Haru Matsumoto

Academic Editor

PLOS ONE

Additional Editor Comments (optional):

Congratualtions! Your paper has significantly improved. I appreciate all the efforts you put in.
---

## [Editor Report · Acceptance letter]

16 Jun 2020

PONE-D-19-33956R2 

Learning to localize sounds in a highly reverberant environment: machine-learning tracking of dolphin whistle-like sounds in a pool 

Dear Dr. Magnasco:

I'm pleased to inform you that your manuscript has been deemed suitable for publication in PLOS ONE. Congratulations! Your manuscript is now with our production department. 

Kind regards, 

on behalf of

Dr. Haru Matsumoto 

Academic Editor

PLOS ONE